

# Eddy-induced Track Reversal and Upper Ocean Physical-Biogeochemical Response of Tropical Cyclone Madi in the Bay of Bengal

Riyanka Roy Chowdhury[1], S. Prasanna Kumar[2], Arun Chakraborty[1]

[1]Centre for Oceans, Rivers, Atmosphere and Land Sciences, Indian Institute of Technology Kharagpur, Kharagpur721302, West Bengal, India

[2]CSIR-National Institute of Oceanography, Dona Paula, Goa 403004, India

*Correspondence to*: S. Prasanna Kumar (prasanna.ocean@gmail.com)

**Abstract.** The life cycle of the tropical cyclone Madi in the southwestern Bay of Bengal (BoB) during 6[th] to 12[th] December 2013 was studied using a suite of ocean and atmospheric data. Madi formed as a depression on 6[th] December and intensified into a very severe cyclonic storm by 8[th] December. What was distinct about Madi was its (1) swift weakening from very severe cyclone to a severe cyclone while moving towards north on 9[th], (2) abrupt track reversal close to 180-degree in a southwestward direction on 10[th], and (3) rapid decay in the open ocean by 12[th] December while still moving southwestward. Using both in situ and remote sensing data, we show that oceanic cyclonic eddies played a leading role in the ensuing series of events that followed its genesis. The sudden weakening of the cyclone before its track reversal was facilitated by an oceanic cyclonic (cold-core) eddy, which reduced the ocean heat content and cooled the upper ocean through upward eddy-pumping of subsurface waters. When Madi moved over the cyclonic eddy-core, its further northward movement was arrested. Subsequently, the prevailing northeasterly winds assisted the slow moving system to change its track to a southwesterly path. While travelling southwestwards, the system rapidly decayed when it passed over cyclonic eddies near the western boundary of the BoB. Though Madi was a category-2 cyclone, it had a profound impact on the physical and biogeochemical state of the upper ocean. Cyclone-induced enhancement in the chlorophyll *a* ranged from 5 to 7-fold, while increase in the net primary productivity ranged from 2.5 to 8-fold. This enhancement of chlorophyll *a* and net primary productivity was much higher than previous cyclones that occurred in the BoB. Similarly, the $CO_2$ out-gassing into the atmosphere showed a 3.7-fold increase compared to the pre-cyclone values. Our study points to the crucial role oceanic eddies play in the life cycle of cyclones and their combined impact on upper-ocean biogeochemical changes in the BoB. Eddies are ubiquitous and tropical cyclones occur in the BoB; there is an urgent need to incorporate eddies in models for better prediction of cyclone track and intensity. As cyclone and eddy co-exists in many parts of the world ocean our approach in delineating the upper-ocean biogeochemical changes can be adapted elsewhere.

## 1 Introduction

The Bay of Bengal (BoB) (Fig. 1) is a tropical sea situated in the eastern part of the northern Indian Ocean. The two most important characteristic features of the BoB are the perennial presence of low salinity waters (30-34 psu) in the upper ocean and the seasonal reversal of atmospheric winds from northeasterly direction between





November and February (5 m/s, northeast or winter monsoon) to southwesterly during June to September (9
m/s, southwest or summer monsoon) (Narvekar and Prasanna Kumar, 2006). This perennial presence of low
salinity waters enhances the stability of the upper water column through increased stratification and makes it
one of the warmest regions in the Indian Ocean. The BoB is a site of tropical cyclones, which occur usually
during pre-monsoon (April-May) and post-monsoon (October-November) periods. Though north Indian Ocean
accounts for only 7% of the total number of tropical cyclones that occur worldwide, the frequency of occurrence
of cyclones in the BoB is 4-times higher than that in the Arabian Sea (Dube et al., 1997). Each year 3-5 cyclones
occur in the BoB, with a primary peak during post-monsoon and a secondary peak in pre-monsoon.
Tropical cyclones in the BoB have been a subject of study by many researchers, which can be broadly classified
into those that deal with (1) prediction of track and intensity of tropical cyclone (e.g., Rao et al., 2007; Basu and
Bhagyalakshmi, 2010; Srinivas et al., 2013; Kanase and Salvekar, 2014; Das et al, 2016; Prakash and Pant,
2017), (2) air-sea interaction and cooling of sea surface temperature (SST) (e.g., O'Brian et al., 1967;
Premkumar et al., 2000; Sadhuram, 2004; Subrahmanyam et al., 2005; Sengupta et al., 2008; McPhaden et al.,
2009; Lin et al., 2009; Kotal et al., 2013; Vissa et al., 2013; Mathew et al., 2018), (3) cyclone-induced
phytoplankton bloom and chlorophyll enhancement (e.g., Madhu et al., 2002; Vinayachandran et al., 2003; Rao
et al., 2006; Patra et al., 2007; Tumula et al., 2009; Sarangi, 2011; Maneesha et al., 2011; Tripathy et al., 2012;
Vidya et al., 2017), and (4) eddy-cyclone interaction (e.g., Ali et al., 2007; Lin et al., 2009;  Sadhuram et al.,
2011; Patnaik et al., 2014).
Though there have been several studies on the tropical cyclone-ocean interaction in the Pacific (typhoon) and
the Atlantic (hurricane) that have advanced our understanding about the upper ocean response in terms of
cooling of SST and enhancement of chlorophyll (e.g., Chang and Anthes, 1978; Price, 1981; Emanuel, 1999;
Babin et al., 2004; Wada and Chan, 2008; Liu et al., 2009; Pun et al., 2011) and cyclone-eddy interaction (e.g.,
Shay et al., 2000; Jaimes and Shay, 2009; Lin et al., 2011; Yablonsky and Ginis, 2013; Sun et al., 2014), the
depth-dependent temperature and chlorophyll response is still poorly understood. It is in this context that the
present paper aims at understanding the (1) ocean-atmosphere condition associated with the evolution of
cyclone Madi, a category-2 cyclone (Saffir-Simpson scale), during December 2013 in the BoB, its sudden
weakening and close to180-degree track reversal before its dissipation, (2) time-evolution of depth-dependent
temperature and chlorophyll profiles in the vicinity of cyclone Madi, and (3) cyclone-induced physical and
biogeochemical response of the upper ocean.
**2 Materials and Methods**
**2.1 Data**
In the present study, the information on cyclone Madi was taken from Indian Meteorological Department (IMD)
(http://www.imd.gov.in), while the track information was taken from Unisys Weather
(http://weather.unisys.com/hurricanes/search). The daily SST data was taken from Tropflux (Praveen Kumar et
al., 2012) (http://www.incois.gov.in/tropflux_datasets/ data/ daily/), while daily sea level anomaly (SLA) along
with zonal and meridional geostrophic current data were taken from AVISO



(https://www.aviso.altimetry.fr/en/my-aviso.html). The zonal and meridional components of wind at 10 m
height were taken from Advanced Scatterometer (ASCAT) level 3 product (Bentamy and Croize-Fillon, 2012)
(https://opendap.jpl.nasa.gov/opendap/OceanWinds/ascat/preview/L2/metop_a/12km/contents.html). It is a daily
product having a spatial resolution of 0.25 degree latitude by longitude. This has been further used for the
calculation of wind stress curl and Ekman pumping velocity (Gill, 1982) as given below:
Wind stress curl $\quad = \quad \frac{\partial \tau_y}{\partial x} - \frac{\partial \tau_x}{\partial y}$ $\qquad\qquad$ (1)
Ekman pumping velocity $\quad = -\frac{1}{\rho f}\left(\frac{\partial \tau_y}{\partial x} - \frac{\partial \tau_x}{\partial y}\right)$ $\qquad\qquad$ (2)
where $\tau_x, \tau_y$ are the zonal and meridional wind stress components, $\rho$ is the density of sea water with its value
taken as 1026 k gm$^{-3}$, and $f$ is the Coriolis parameter which varies with latitude.
The oceanic heat content (OHC) in the upper 300 m is calculated following Eq. (3):
$\text{OHC} = \rho c_p \int_{h_2}^{h_1} T(z)\,dz$ $\qquad\qquad$ (3)
where, $\rho$ is the density of seawater, $c_p$ is the specific heat capacity of sea water taken as 3.87 kJ kg$^{-1}$ K$^{-1}$, h$_1$ and
h$_2$ are the lower and upper water depths, and $T(z)$ is the temperature profile measured in Kelvin.
The relative humidity at 500 hpa was taken from NCEP 2 reanalysis daily data having 2.5 degree grid resolution
(https://www.esrl.noaa.gov/psd/data/gridded/data.ncep.html) (Kalnay et al., 1996). The daily zonal and
meridional components of wind at 850, 500 and 200 hpa having a spatial resolution of 0.5 degree were extracted
from NCEP climate forecast system version 2 and used to compute vector wind. Winds at 850 and 200 hpa were
used for the calculation of vertical wind shear (Saha et al., 2014) (http://www.ncep.noaa.gov).
In order to gain insight about the time evolution of temperature and chlorophyll in the upper water column in
response to the passage of cyclone Madi, we have analyzed the trajectory of two Argo floats (WMO ID
2901288, 2901629) for temperature profiles that were in the vicinity of Track 2 and one Bio-Argo float (WMO
ID 2902086) for chlorophyll profiles that was to the right of Track 1 as shown in Fig. 1. Argo/Bio-Argo data
were downloaded from Argo CORIOLIS site (http://www.coriolis.eu.org/Data-Products/Data-Delivery/Data-
selection).
**2.2 Data Processing method of Chlorophyll and net primary production**
The satellite-derived daily chlorophyll *a* (Chl-*a*) pigment concentration data and net primary production (NPP)
estimated based on vertically generalized productivity model (VGPM) of (Behrenfeld and Falkoswski, 1997)
were taken from Moderate Resolution Imaging Spectro-radiometer (MODIS) Aqua Ocean color
(https://oceandata.sci.gsfc. nasa.gov/MODISA/). The Level 3 Chl-*a* dataset has a zonal and meridional
resolution of 0.05 degree longitude by latitude. From the daily data, weekly composites were calculated.



In order to determine the net $CO_2$ flux over southwestern BoB, before, during, and after the passage of the
cyclone Madi, $pCO_2{}^{air}$ data was taken from NOAA ESRL (ftp://aftp.cmd1.noaa.gov/product/
trends/co2/co2_mm_g1.txt). Since the daily $pCO_2{}^{sea}$ values are not available, the value of climatological air-sea
difference in partial pressure of $CO_2$ was taken from (Takahashi et al., 2009) and the net flux was calculated
using the following formula:
$$F = k.a.(pCO_2{}^{sea} - pCO_2{}^{air}) \qquad (4)$$
where, $k$ denotes the gas transfer velocity, $a$ is the solubility of $CO_2$ in sea water which is dependent on sea
surface temperature and salinity (Weiss, 1974) as per the following equations:
$$\ln a = A_1 + A_2 \left(\frac{100}{T}\right) + A_3 \ln \left(\frac{T}{100}\right) + S \left[B_1 + B_2 \left(\frac{T}{100}\right) + B_3 \left(\frac{T}{100}\right)^2\right] \qquad (5)$$
The gas-transfer velocity "k" is calculated using wind speed following (Wanninkhof, 1992) by using the
formula
$$k(cm\ h^{-1}) = \Gamma \cup^2 \left(\frac{S_c}{660}\right)^{-1/2} \qquad (6)$$
where $\Gamma$ is the scaling factor and its value of 0.26 is taken from (Takashashi et al., 2009), while $\cup$ is the wind
speed. $S_c$ is the Schimidt number (kinematic viscosity of water/diffusion coefficient of $CO_2$ in water), the value
of which is 660 for $CO_2$ in seawater at 20°C and is a function of temperature and is computed as:
$$S_c = A - BT + CT^2 - DT^3 \qquad (7)$$
For the values of the constants A, B, C and D refer (Weiss, 1974; Wanninkhof, 1992).
We have divided the study region into Box A, Box B, Track 1, Track 2 and Box abcd. See Fig.1 for the location
of these sub-regions.

## 2.3 Origin, evolution and decay of the cyclone Madi

On 30[th] November 2013, as per the Indian Daily Weather Report of India Meteorological Department (IMD), a
low pressure system was formed over the southwestern part of the BoB (Fig. 1) and slowly intensified into a
depression (the classification of intensity of the system is based on IMD,
http://imd.gov.in/section/nhac/termglossary.pdf) on 6[th] December 2013 with its centre at 10°N and 84°E (Fig. 1).
The system intensified further into a deep depression (DD) on the same day with maximum sustained wind
speed of 50-60 km/hr. Subsequently, when it turned into a cyclonic storm (CS) on 7[th] December, the IMD
named it as Madi. On further intensification into a severe cyclonic storm (SCS), the system started moving in a
north/north-northeast direction with maximum sustained wind speed of 90-100 km/hr. Subsequently, on 8[th]
December, the system turned into a very severe cyclone (VSCS) with maximum sustained wind speed of 120-
130 km/hr. The system moved further northward on 9[th] December reaching the location 14.6°N and 84.7°E,
when it weakened into SCS with maximum sustained wind speed of 110-120 km/hr. The system not only



weakened but slowed down considerably while reaching the location 15.7°N and 85.3°E on 10th December
where it remained stationary for a while. At that point the SCS deviated from its northward track, took a near
180 degree turn and veered southwestward (Fig. 1). During the course of its south-westward movement, the SCS
weakened to CS with maximum sustained wind speed of 80-90 km/hr. On its further south-westward journey,
the CS weakened to DD on 11th December and further to a depression the same day with its centre at 12.9°N and
82.7°E. On 12th December 2013 the depression further weakened to a well marked low pressure.
**3 Results and Discussion**
We start our analysis by examining the time-evolution of the spatial distribution of various oceanic and
atmospheric parameters from 4th to 15th December 2013 to understand the thermo-dynamical and dynamical
conditions that led to the formation and subsequent dissipation of cyclone Madi.
**3.1 Thermodynamic conditions before, during and after the cyclone**
Ocean heat content (OHC) plays an important role in the translation speed and intensification of cyclones over
the BoB (Sadhuram et al., 2010). The time-evolution of the spatial distribution of the OHC on 4th December
2013 showed large values ranging from 3.580 to 3.600 x $10^{11}$ J/m², except a meridionally-elongated region
along the western boundary between 8° and 20°N, and another small patch in the central BoB centered at 13°N,
where the values were small (Fig. 2a, 2b). Note the meridional band of the large OHC, adjacent to the
meridional band of small OHC hugging the western boundary, with three distinct patches of high values within
them. The drastic decrease of OHC on 7th December (Fig. 2c, 2d) indicated strong heat uptake by the cyclonic
storm during the process of its intensification. As the system moves northward, passing over the region of high
OHC it continues to take up heat from the upper ocean and intensifies further (Fig. 2e). Note that on 9th
December when the track of the system passes over a region of low OHC it weakens (Fig. 2f). On 10th
December, when the system it deviated from its northward track and took almost a 180-degree turn it was
passing through low OHC (Fig.2g). On its southward journey, the system passes over regions of lower OHC on
11th and 12th (Fig. 2h, 2i), when it dissipates into DD and to well-marked low pressure respectively. Once the
system is dissipated, the spatial distribution of OHC showed a recovery in terms of heat gain by the upper ocean
(Fig. 2j-l), especially in the region of the track of the cyclone.
**3.2 Dynamic conditions before, during and after the cyclone**
The analysis of the time-evolution of the spatial distribution of sea level anomaly (SLA) over-laid with
geostrophic current from 4th to 15th December revealed the presence of several meso-scale cyclonic (blue region
with negative SLA) and anticyclonic (red regions with positive SLA) eddies (Fig. 3). The SLA and associated
geostrophic current clearly indicated the presence of two cyclonic eddies along the western boundary and two in
the offshore region (Fig. 3a, b). The region of occurrence of these cyclonic eddies coincided with the region of
low OHC (Fig. 2). Note that the genesis of Madi in the form of a depression occurred on 6th December in the
region of positive SLA with an anticyclonic circulation (Fig. 3c), which was the same region that had high




OHC. The intensification of Madi on 8$^{th}$ December also occurred in a region of positive SLA with an
anticyclonic circulation (Fig. 3d, 3e). On 9$^{th}$ December when the system entered into a region of negative SLA
with cyclonic circulation (Fig. 4f), which was also a region of low OHC it weakened as it was deprived of the
thermal energy from the upper warm ocean that is essential to sustain the system. On 10$^{th}$ December when the
system moved further north entering towards the core of the cyclonic eddy (Fig. 3g) with low OHC (Fig. 2g) its
further northward movement was arrested. It remained stationary for a while and changed its track to almost
180-degree in a southwestward direction. While doing so the cyclone Madi was moving further through the
regions of strong cyclonic circulation/eddies (Fig. 3i), which rapidly reduced its strength and finally led to its
dissipation on 12$^{th}$ December. The passage of cyclone Madi modified the upper ocean circulation in the
southwestern part of the BoB (Fig. 3j-l) into a large cyclonic gyre with strong southward western boundary
current from 17$^{o}$ to 10$^{o}$N along the west coast of India. The four cyclonic eddies were now prominently seen
embedded in this large-scale gyre.
When a cyclone passes over the cyclonic eddy region, the colder temperature within the eddy could potentially
reduce the translation speed of the cyclone as well as its intensity as it is unable to fuel the cyclone as effectively
as in the case of the warm water region where it originates. In order to further ascertain the role of cyclonic eddy
in weakening the strength of the cyclone before its track reversal, we calculated the translational speed of the
system from its formation on 6$^{th}$ to its dissipation on 12$^{th}$ December and examined it along with its strength
(Table 1). It is clear from Table 1 that on 9$^{th}$ December when the cyclone entered the region of oceanic cyclonic
eddy the translational speed of the cyclone decreased from 2.81 m/s to 1.96 m/s and the cyclone weakened from
VSCS to SCS. Thereafter, subsequent to track reversal as the system moves south-westward out of the cyclonic
eddy, the translation speed increases.
Though the weakening and the final dissipation of the cyclone Madi was easy to understand in the context of the
prevailing oceanic cyclonic eddies, we examined the time-evolution of the spatial distribution of the
atmospheric parameters such as wind at 850 hpa (Fig. 4), vertical wind shear between the 850 and 200 hPa (Fig.
5) and mid-troposheric (500hpa) relative humidity (Fig. 6) to understand the atmospheric condition.
The salient feature of the large-scale atmospheric circulation over the BoB, prior to the genesis of cyclone Madi,
was the prevalence of an easterly zonal wind with speed between 5 and 15 m/s with an embedded cyclonic
circulation located in the southwestern region (Fig. 4a, 4b). The wind speed associated with the cyclonic
circulation was between 15 and 25 m/s. On 6$^{th}$ December when the depression was formed, this broad cyclonic
circulation becomes well organized with a small central region having lower wind speeds of 10 m/s, while the
surrounding regions had higher wind speeds of 20-25 m/s (Fig. 4c). When the system developed into the CS
(Fig. 4d) and intensified into a VSCS (Fig. 4e), the large-scale atmospheric circulation in the BoB showed a well
defined "eye of the cyclone". Away from the cyclonic circulation, the winds in the northen part of the BoB were
mostly southwestward. On 9$^{th}$ December the weakening of the system was discernible as it moved northward
(Fig. 4f). At this time the low vertical wind shear (10 to 15 m/s) (Fig. 5f) and high relative humidity (60-80 %)
(Fig. 6f) were congenial for the system for further intensification or at least to sustain its intensity. In contrast
the system weakened from VSCS to SCS. This indicated that the system evolution at this time was controlled by
the oceanic cyclonic eddies rather than the atmospheric conditions. On 10$^{th}$ when the system reached its northern


most location (Fig. 4g), it was actually sitting right on the top of the cold-core of the cyclonic eddy (Fig. 3g). At
this time the system became stationary and the prevailing easterly winds (Fig. 4g) were able to turn and move it
towards southwesterly direction, a result which is consistent with that of (Bhattacharya et al., 2015).
Thus, our study showed that the weakening of cyclone on its northward journey was mediated by the oceanic
cold-core cyclonic eddy while the change in the direction of the cyclone track when the system was stationary
was brought about by the prevailing northeasterly winds.
### 3.3 Cyclone-induced along track oceanic variability
In order to quantify the upper ocean response of the tropical cyclone Madi, we examined four oceanic
parameters viz. SST, Ekman pumping velocity (EKV), SLA and OHC during the period 2 to 15 December 2013
at four locations : (1) Box A, the region of genesis of the depression which subsequently turned into cyclone
Madi, (2) along Track 1, the northward path followed by the cyclone Madi during which time it intensified from
CS to VSCS, (3) Box B, the region where the cyclone Madi weakened, remained stationary and eventually
turned, and (4) along Track 2, the southwestward path of the cyclone which eventually dissipated.
The time-evolution of SST in Box A, showed a monotonic decline of $1.5^oC$ from $28.2^oC$ to $26.7^oC$ during the
period from the genesis of the cyclone to its decay (Fig. 7). However, the rate of decrease during the entire
period was not uniform. Even before the formation of the depression SST showed a weak decrease of $0.3^oC$,
however, during the period 6[th] to 8[th] December when the depression was formed within the Box A and turned
into a cyclone, the SST decreased rapidly. Though the system was away from the region of Box A and was
dissipating with time during 9[th] to 11[th] December, the SST within the Box A showed the most rapid decrease of
$1.1^oC$. The SLA, on the other hand, showed a continuous decrease, before the formation of depression and much
after its dissipation. The SST showed a recovery/warming trend after 12[th] December. The EKV showed a peak
on 6[th] December, at the time of formation of depression. This is expected, as under the action of cyclonic wind,
the upward Ekman pumping will also increase in magnitude. What was unexpected was the temporal response
of the OHC, which showed an initial decrease from 2[nd] to 3[rd] December followed by an increase reaching the
highest value of $3.589 \times 10^{11}$ J/m$^2$ and a subsequent decrease.  A secondary peak occurred on 6[th] as the
depression formed in the area of Box A. During 6[th] to 8[th] December when the system intensified and was located
within the Box A, the OHC showed rapid decrease to a value of $3.574 \times 10^{11}$ J/m$^2$. There after the values were
closer to $3.576 \times 10^{11}$ J/m$^2$, except on 13[th] December when it once again peaked to $3.578 \times 10^{11}$ J/m$^2$.
Though the response of all the four parameters along Track 1 (Fig. 8), Track 2 (Fig. 9) and at Box B (Fig. 10)
were similar to that of Box A (Fig. 7), a closer similarity was noticed between Box A and Track 2, and between
Track 1 and Box B. However, the magnitudes of response of each parameter and their times of occurrence were
different depending on the position of the cyclone with respect to each of the four locations.  For example, the
OHC showed an inverse relationship with the Ekman pumping velocity along Track 1(Fig. 8) and at Box B (Fig.
9), while at Box A (Fig. 7) and along Track 2 (Fig. 8) the OHC showed a double-peak structure. Along Track 1,
the occurrence of highest value of EKV was consistent with the system intensifying into VSCS with a maximum
sustained wind speed of 110-120 km/hr. Similarly, at Box B also the occurrence of the highest EKV coincided



with the arrival of the cyclone at this location. The rapid decrease in SST occurred in all the four regions, in
general, during 9th to 11th December, indicating a time-lag between the presence of the cyclonic storm and the
peak of the upward EKV. Another noteworthy feature, common in all the four cases, was the co-variation of
SST and SLA, both showing a monotonic decline, indicating the occurrence of colder waters associated with
decreasing sea level, except along Track 2 (Fig. 9). Note that this lowered sea level and colder SST occurred
well before the initiation of the upward Ekman pumping under the influence of the cyclone Madi. This pointed
towards the pre-cyclone cooling of SST by oceanic cyclonic eddies, which was also evident from the time
evolution of the spatial maps of daily SLA (Fig. 3). However, along Track 2, SLA showed a rapid increase from
2nd to 5th December followed by a slower increase until 8th December, well before the passage of the cyclone
through this region. This is primarily due to the fact that the location of Track 2 passes through an anticyclonic
eddy.
**3.4 Depth-dependent temperature and chlorophyll _a_ response**
Having analyzed the cyclone-induced SST response along the track of the cyclone, it is pertinent to examine the
vertical profiles of temperature before, during and after the passage of cyclone Madi. Hence, we examined the
vertical profiles of temperature in the vicinity of Track 2 obtained by two Argo floats (ID-2901288 and ID-
2901629) which transected the northern and southern parts of Track 2 during the period of study (see Fig. 1 for
the location of Argo floats). The vertical profiles of temperature obtained from both the Argo floats (Fig. 11a, b)
showed the presence of a thermal inversion (0.2 to 0.3°C) located in the upper 40 m prior to the passage of the
cyclone Madi, which disappears in the subsequent profiles.  The most distinct change was in the mixed layer
temperature and depth. On 4th December prior to the formation of cyclone the mixed layer depth (MLD)
obtained from Argo float with ID-2901288 was 30m and temperature was 28.2°C and after the passage of
cyclone on 14th December the MLD was 50 m and temperature was 26.5°C (Fig. 11a). A similar change was
also noticed in the vertical profiles of temperature obtained from Argo float with ID-2901629 (Fig. 11b). Thus,
both the Argo floats captured the cyclone-induced mixed layer cooling and deepening.
The vertical profiles of Chl-_a_ obtained from the Bio-Argo float (ID-2902086) showed low values prior to the
cyclone (23rd November to 3rd December 2013) in the range of 0.10 to 0.15 mg/m$^3$ with constant value within
the mixed layer and a subsurface chlorophyll maximum (SCM) located at  about 50m (Fig. 11c). The vertical
profiles of Chl-_a_ showed a progressive increase during and after the cyclone in both the surface as well as the
subsurface values reaching a maximum of 0.45 and 0.65 mg/m$^3$ respectively on 23rd December 2013. Thereafter,
it showed a decline on 28th December 2013 when the value in the upper 60 m was 0.40 mg/m$^3$ with no
perceptible SCM. Thus, the Chl-_a_ profiles in the upper 60 m showed maximum impact due to the cyclone
leading to an overall increase in the biomass.



### 3.5 Cyclone-induced biogeochemical variability

It is well known that tropical cyclones bring about large changes in the upper ocean productivity as well as gas-exchange between ocean and atmosphere. In order to understand and quantify the biogeochemical response due to the cyclone Madi, we examined along track variation of satellite-derived chlorophyll $a$ pigment concentration (Chl-$a$), net primary production (NPP), and the net $CO_2$ flux. A major difficulty with the remotely sensed Chl-$a$ pigment concentration is the lack of adequate cloud-free pixels along track on a daily time scale. In order to overcome this, we have used weekly composite data for Chl-$a$ for the calculation of NPP from 30$^{th}$ November to 28$^{th}$ December in the four regions, viz. Box A and B and Track 1 and 2 (Fig. 1), while the net $CO_2$ flux was computed on daily time scale from 2$^{nd}$ to 15$^{th}$ December 2013.

The time variation of the weekly composite of Chl-$a$ showed a pattern that was typical of the cyclone induced response (Fig. 12). Prior to the genesis of cyclone Madi, the Chl-$a$ was in the range of 0.2 to 0.4 mg/m$^3$, but the weekly composite values for the period 7$^{th}$ to 14$^{th}$ December, which includes the growth, decay and a couple of days after cyclone, showed several fold increase. The maximum increase of 2.7 mg/m$^3$ was in Box B, which was almost 7-times higher than the pre-cyclone period. The minimum increase of 1 mg/m$^3$ occurred along the Track 2, which was 5-times higher than the pre-cyclone period. In the Box A and along Track 1, the Chl-$a$ values were 1.4 and 1.5 mg/m$^3$ respectively after the cyclone. It is pertinent to examine the chlorophyll enhancement by other cyclones in the BoB and compare with the present study. For example, the Orissa super cyclone in October 1999 produced a Chl-$a$ enhancement in the rage of 0.38 to 0.97 mg/m$^3$ in the open ocean region (Madhu et al., 2002), while that near the land fall region was a maximum of 1.0 mg/m$^3$ (Patra et al., 2007). (Vinayachandran et al., 2003) reported a value ranging from 0.5 to 2.0 mg/m$^3$ for the cyclones that occurred during November-December during the period 1996 to 2001. In the case of cyclone Sidr in 2007, (Maneesha et al., 2011) obtained an increase from 0.2 to 0.5 mg/m$^3$.

Thus, the Chl-$a$ enhancement by Madi was much greater than for previous cyclones that occurred in the BoB. The obvious question would be why the Chl-$a$ along both the tracks as well as the boxes showed an increase and why Box B showed the highest magnitude of Chl-$a$ response to the cyclone. Recall that the EKV showed a rapid increase during the period when the cyclone was transiting these regions, while a concomitant rapid decrease of SST was also noticed. This indicated the upward transport of cold subsurface waters under the influence of the cyclonic winds. As the subsurface waters are nutrient rich, the increased Ekman pumping under the tropical cyclone would bring more nutrients to the upper oceans which will kick-start the photosynthesis (Subrahmanyam et al., 2002; Lin et al., 2003) resulting in the observed increase in the Chl-$a$ biomass. Recall also that an oceanic cyclonic eddy was located in the region of Box B where the cyclone was stationary for a while. In the BoB, the nutricline is located just below the mixed layer, usually at a depth ranging from 20 to 40m (Prasanna Kumar et al., 2007), and the eddy-pumping (Falkowski et al., 1991) associated with oceanic cyclonic eddies is able to supply sub-surface nutrients to the surface waters (Prasanna Kumar et al., 2004). Hence, we infer that under the combined effect of the oceanic eddy and the cyclone Madi, the upward Ekman pumping would have been stronger and more nutrients could be supplied to the upper ocean, which resulted in the observed 7-fold increase. The lowest response, 5-fold increase, was seen along track 2, which is to be expected as when the cyclone transited along this path it was decaying rapidly. Note that by the last week of



December (21-28) the Chl-*a* values came back to their pre-cyclone values. Thus, in response to the cyclone Madi, the Chl-*a* in all the 4 regions, which were under its influence, exhibited enhancement, though to varying magnitudes. The increase in Chl-*a* concentration was rapid during the enhancement period, while the decline took more time.

Consistent with the Chl-*a* response, the NPP (Fig.13) showed a similar pattern of co-variability with highest value of 2500 mg C $m^{-2}$ $day^{-1}$ occurring in Box B, which was also 8-fold higher than the pre-cyclone value. Similarly, the least enhancement in NPP was shown along Track 2 with a value of 800 mg C $m^{-2}$ $day^{-1}$, which was only 2-and-half fold increase from its pre-cyclone value. The enhancements along Track1 and at Box A were 5-fold and 2-fold respectively compared to pre-cyclone values. Being a weekly composite, in both Chl-*a* and NPP, it was not possible to resolve the exact date of enhancement or decline, though the overall pattern was discernible.

It has been shown by several studies that out-gassing of $CO_2$ from ocean to atmosphere takes place under the influence of tropical cyclones (see for e.g. Bates et. al., 1998; Nemeto et al., 2009). This happens in two ways – firstly, the strong wind associated with cyclones results in out-gassing from ocean to atmosphere; secondly, the supply of subsurface dissolved inorganic carbon to the surface due to the upward Ekman pumping by the wind stress curl and it's out-gassing due to heating and equilibration with the atmosphere. We examined the daily variation of total $CO_2$ flux in all the 4 regions from 2nd to 15th December 2013 (Fig. 14) to decipher this. All the four regions showed enhanced net $CO_2$ flux to the atmosphere, though the magnitude and timing were different. Again consistent with Chl-*a* and NPP, the maximum $CO_2$ out-gassing to the atmosphere was seen in Box B and the least was along Track 2. In Box B, the fastest $CO_2$ out-gassing of 4.7 Tg carbon per day to the atmosphere took place when the cyclone Madi was in this Box region and was 3.7-fold higher than its average pre-cyclone value. The maximum $CO_2$ out-gassing of 2.5 Tg carbon per day to the atmosphere took place along Track 2 and was 2-fold greater than its average pre-cyclone value. The maximum value of $CO_2$ out-gassing at Box A and along Track 1 was similar to that of Track 2. A secondary peak in $CO_2$ out-gassing was seen at Box B and along Track 2 on 11th December 2013, while Box A and Track 1 did not show such a pattern.

## 4. Summary and Concluding Remarks

The ocean-atmosphere conditions associated with category-2 tropical cyclone Madi in the southwestern BoB during 6th to 12th December 2013 were studied using a suite of in situ and remote sensing data sets. We infer that the origin of cyclone Madi and its and strengthening from CS to VSCS was facilitated by the large OHC. On its northward movement when it passed over an oceanic cold-core cyclonic eddy, the system weakened to SCS and its translation speed was decreased by almost 1 m/s. In spite of the prevailing favorable atmospheric conditions for the strengthening of a cyclone, such as low vertical wind shear and high relative humidity, the system did not strengthen further; instead it remained weak. At this stage the prevailing northeasterly winds altered the track of the weakened system by almost 180-degree. On its southward journey the system passed over cold-core eddies that rapidly dissipated it.





The cyclone Madi triggered intense physical and biogeochemical response in the upper ocean. The weekly
composite of satellite-derived Chl-*a* pigment concentration showed an enhancement that ranged from 5 to 7-fold
with a maximum value of 2.7 mg/m$^3$. A similar response was seen in the net primary productivity which showed
a 2.5 to 8-fold increase, with a maximum value of 2500 mg C m$^{-2}$ day$^{-1}$. The largest values of both Chl-*a* and
NPP was greater than for previous cyclones in the BoB. Out study indicates that a combination of an oceanic
cyclonic eddy along with cyclone Madi facilitated upward Ekman pumping of nutrient rich subsurface waters to
the surface, thereby kick-starting the primary production and increasing the chlorophyll biomass. Consistent
with this, the net $CO_2$ out-gassing to the atmosphere also was the greatest in this region amounting to 4.7 Tg
carbon per day, which was 3.7-fold greater than the pre-cyclone values. Our study emphasizes the importance of
eddy-cyclone interaction that led to the large increase in Chl-*a*, primary production and $CO_2$ out-gassing. Since
cyclone and eddies co-occur in many parts of the world ocean our approach can be adopted in other regions to
quantify the biogeochemical response.
One of the limitations of our study is the lack of modeling to quantify the eddy-cyclone interaction. Our study
underscores the important role of oceanic eddies in understanding the life cycle of tropical cyclones in the BoB.
Since both cyclonic and anticyclonic eddies are ubiquitous in the BoB, they will impact both the translation
speed and intensity of a tropical cyclone. Hence, for the accurate prediction of a cyclone track and its intensity,
there is an urgent need to incorporate eddies into the predictive models; this action is still to be explored. This
will be attempted in the near future.
**Author Contribution**
S. Prasanna Kumar and Arun Chakraborty formulated the problem, and were involved continuously during the
conduction of the work. Riyanka Roy Chowdhury has been the primary researcher and has carried out all the
data analyses, computation and derivations, and preparation of the graphics required for the manuscript. S.
Prasanna Kumar and Riyanka Roy Chowdhury were involved in the preparation of the manuscript.
**Acknowledgements**
The authors thank Directors of Indian Institute of Technology (IIT), Kharagpur and CSIR-National Institute of
Oceanography (CSIR-NIO), Goa and the Council of Scientific and Industrial Research (CSIR), New Delhi for
all the support and encouragement for this research. The daily SST data are available from
(http://www.incois.gov.in/tropflux_datasets/ data/ daily/), daily SLA data along with zonal and meridional
geostrophic velocity are from  (https://www.aviso.altimetry.fr/en/my-aviso.html), surface wind data are from
(https://opendap.jpl.nasa.gov/opendap/OceanWinds/ascat/preview/L2/metop_a/12km/contents.html), while the
wind at 850 hpa is from(http://www.ncep.noaa.gov). The Argo data are from (http://www.coriolis.eu.org/Data-
Products/Data-Delivery/Data-selection). The graphics were generated using MATLAB and the code used in this
paper can be obtained from the first author. Riyanka Roy Chowdhury acknowledges Ministry of Human
Resource Development for providing the research fellowship. NIO contribution number is XXXX.



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




**Table 1** Translation speed of the cyclonic disturbance along with its category during the life cycle of
cyclone Madi. L-low pressure, DD-deep depression, CS-cyclonic storm, SCS-severe cyclonic storm, VSCS-
very severe cyclonic storm.

| Date | Translation Speed (m/s) | Category of Cyclonic Disturbance |
|---|---|---|
| 06/12/2013 | 1.63 | DD |
| 07/12/2013 | 2.48 | CS to SCS |
| 08/12/2013 | 2.81 | VSCS |
| 09/12/2013 | 1.96 | SCS |
| 10/12/2013 | 2.32 | CS |
| 11/12/2013 | 3.44 | DD |
| 12/12/2013 | 5.41 | L |




















**Figure Captions**

**Figure 1** Map showing the track of the tropical cyclone Madi (magenta filled circles inside the black
circles) during 6-12 December 2013 in the Bay of Bengal obtained from UNISYS Weather.  The shading is
the sea level anomaly (m), while vectors are the wind (m/s) at 850 hpa, both are composite for the period
6-12 December 2013. Location of Box A, Track 1, Box B, Track 2, rectangular Box abcd, and Argo floats
(ID-2901288 red plus & ID-2901629 yellow plus) near Track 2 are also shown in the map. The black
hollow circles (seen as dark circles due to overlap) show the position of Bio-Argo float (ID2902086).
**Figure 2** Spatial maps of oceanic heat content ($\times 10^{11}$ J/m$^2$) from 4$^{th}$ (a) to 15$^{th}$ (l) December 2013 with track
of the cyclone overlaid. The black filled circles represent the position of the cyclone on a particular day,
while the magenta filled circles indicate the track.
**Figure 3** Spatial maps of sea level anomaly (m) from 4$^{th}$ (a) to 15$^{th}$ (l) December 2013 with track of the
cyclone overlaid. The black filled circles represent the position of the cyclone on a particular day, while the
magenta filled circles indicate the track.
**Figure 4** Spatial maps of wind speed (shading, m/s) overlaid with wind vectors (thin arrow) at 850 hpa
from 4$^{th}$ (a) to 15$^{th}$ (l) December 2013 with track of the cyclone overlaid. The black filled circles represent
the position of the cyclone on a particular day, while the magenta filled circles indicate the track.
**Figure 5** Spatial maps of vertical wind velocity difference between the 850 and 200hPa (shading, m/s) from
4$^{th}$ (a) to 15$^{th}$ (l) December 2013 with track of the cyclone overlaid. The black filled circles represent the
position of the cyclone on a particular day, while the magenta filled circles indicate the track.
**Figure 6** Spatial maps of relative humidity (%) overlaid with winds at mid-troposhere (500hpa) from 4$^{th}$ (a)
to 15$^{th}$ (l) December 2013 with track of the cyclone overlaid. The black filled circles represent the position
of the cyclone on a particular day, while the magenta filled circles indicate the track.
**Figure 7** Space-averaged variation of the sea surface temperature (SST, $^o$C), Ekman pumping velocity
(EKV, m/day, positive upward), oceanic heat content (OHC, x $10^{11}$ J/m$^2$) and sea level anomaly (SLA, m)
in Box A from 2-15 December 2013.
**Figure 8** Along track variation of  the sea surface temperature (SST, $^o$C), Ekman pumping velocity (EKV,
m/day, positive upward), oceanic heat content (OHC, x $10^{11}$ J/m$^2$) and sea level anomaly (SLA, m) along
Track 1 from 2-15 December 2013. These are daily averages along the track.
**Figure 9** Space-averaged variation of the sea surface temperature (SST, $^o$C), Ekman pumping velocity
(EKV, m/day, positive upward), oceanic heat content (OHC, x $10^{11}$ J/m$^2$) and sea level anomaly (SLA, m)
in Box B from 2-15 December 2013.





**Figure 10** Along track variation of the sea surface temperature (SST, $^{o}$C), Ekman pumping velocity (EKV,
m/day, positive upward), oceanic heat content (OHC, x $10^{11}$ J/m$^2$) and sea level anomaly (SLA, m) along
Track 2 from 2-15 December 2013. These are daily averages along the track.
**Figure 11** Time-series of the vertical profiles of temperature ($^{o}$C) in the vicinity of Track 2 obtained from
(a) Argo float ID-2901288 for 4, 9, 14, 19 and 24 December 2013, (b) Argo float ID-2901629 for 2, 12 and
22 December 2013 and (c) chlorophyll $a$ (mg/m$^3$) in the vicinity of Track 1 obtained from Bio-Argo ID-
2902086 for 23 and 28 November and 3, 8, 13, 18, 23 and 28 December 2013.
**Figure 12** Time variation of weekly composite of chlorophyll $a$ pigment concentrations (Chl-$a$, mg/m$^3$) in
the Box A (red) and B (blue) and along Track 1 (green) and 2 (black) from 30 November to 28 December
2013. The vertical lines are the standard deviations.

**Figure 13** Time variation of weekly composite of net primary production (NPP, mg C m$^{-2}$ day$^{-1}$) in the Box
A (red) and B (blue) and along Track 1 (green) and 2 (black) from 30 November to 28 December 2013. The
vertical lines are the standard deviations.
**Figure 14** Daily variation total CO2 flux (terra gram carbon per day) in the Box A (red) and B (blue) and
along Track 1 (green) and 2 (black) from 2 to 15 December 2013. The vertical lines are the standard
deviations.

















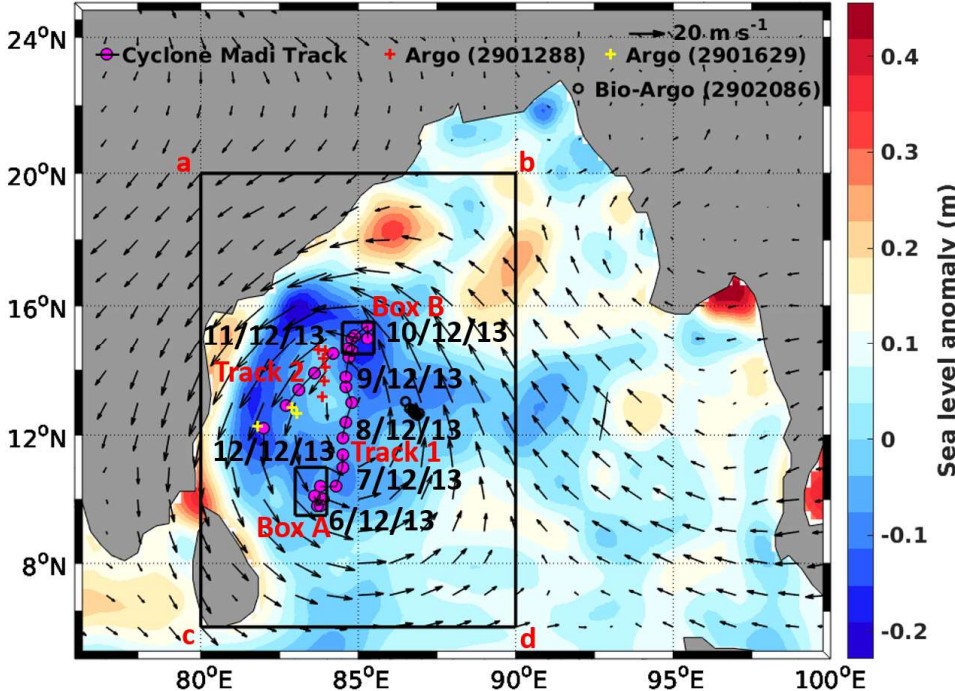

**Figure 1** Map showing the track of the tropical cyclone Madi (magenta filled circles
inside the black circles) during 6-12 December 2013 in the Bay of Bengal obtained from
UNISYS Weather. The shading is the sea level anomaly (m), while vectors are the wind
(m/s) at 850 hpa, both are composite for the period 6-12 December 2013. Location of Box
A, Track 1, Box B, Track 2, rectangular Box abcd, and Argo floats (ID-2901288 red plus
& ID-2901629 yellow plus) near Track 2 are also shown in the map. The black hollow
circles (seen as dark circle due to overlap) show the position of Bio-Argo float
(ID2902086).


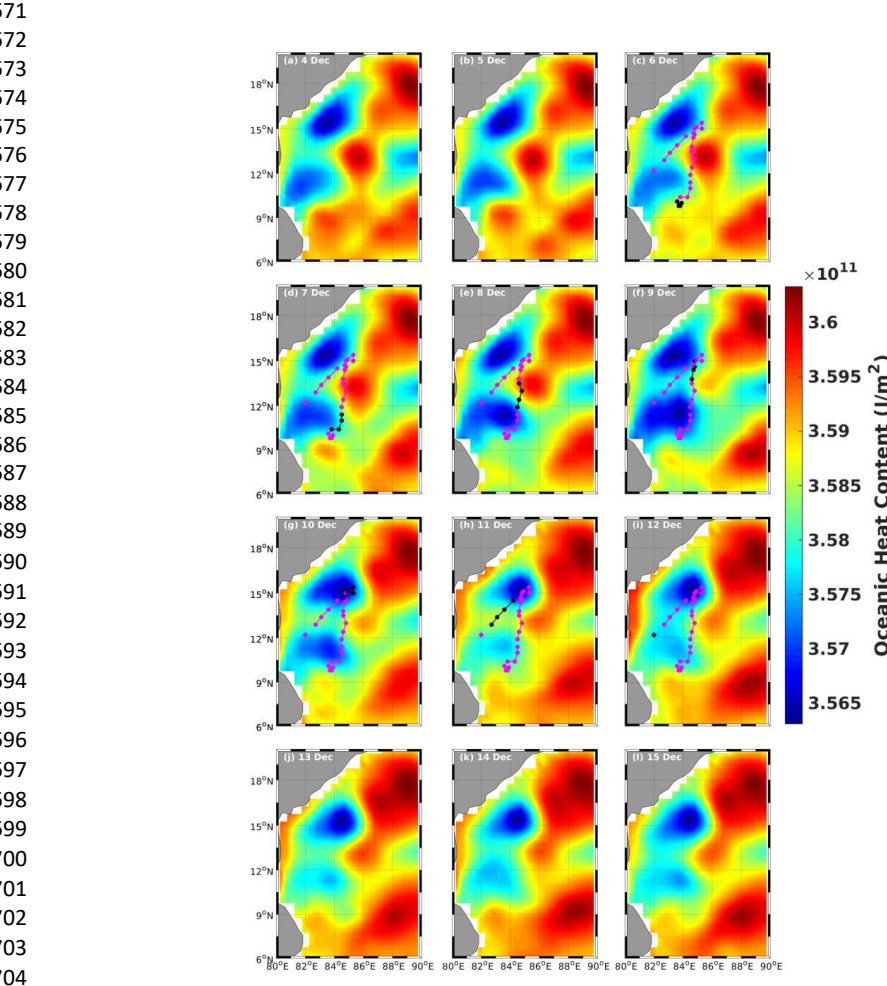

**Figure 2** Spatial maps of oceanic heat content ($\times 10^{11}$ J/m$^2$) from 4$^{th}$ (a) to 15$^{th}$ (l)
December 2013 with track of the cyclone overlaid. The black filled circles represent the
position of the cyclone on a particular day, while the magenta filled circles indicate the
track.



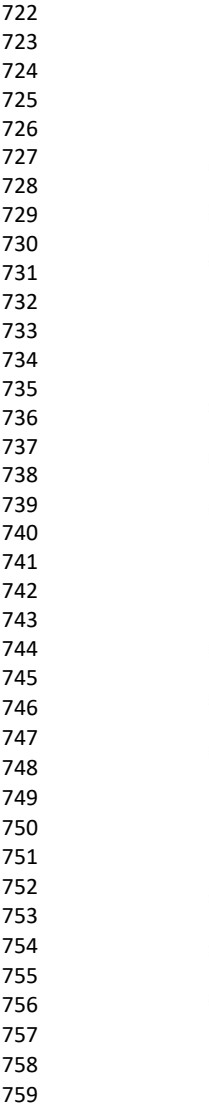

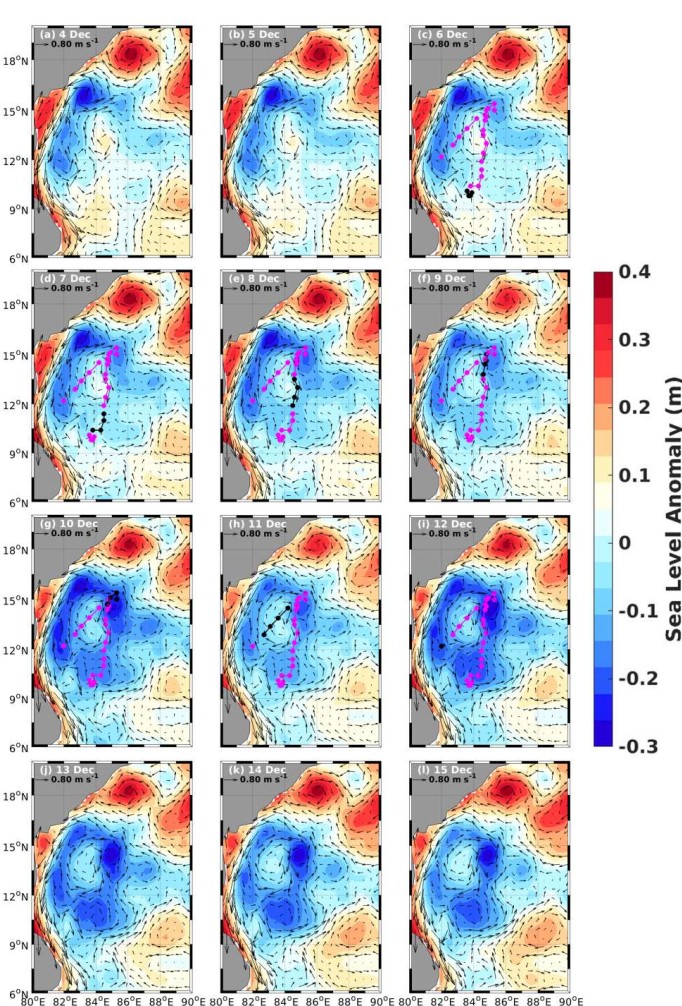

**Figure 3** Spatial maps of sea level anomaly (m) from 4[th] (a) to 15[th] (l) December 2013
with track of the cyclone overlaid. The black filled circles represent the position of the
cyclone on a particular day, while the magenta filled circles indicate the track.

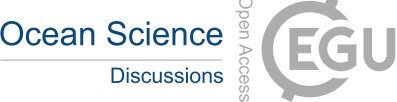


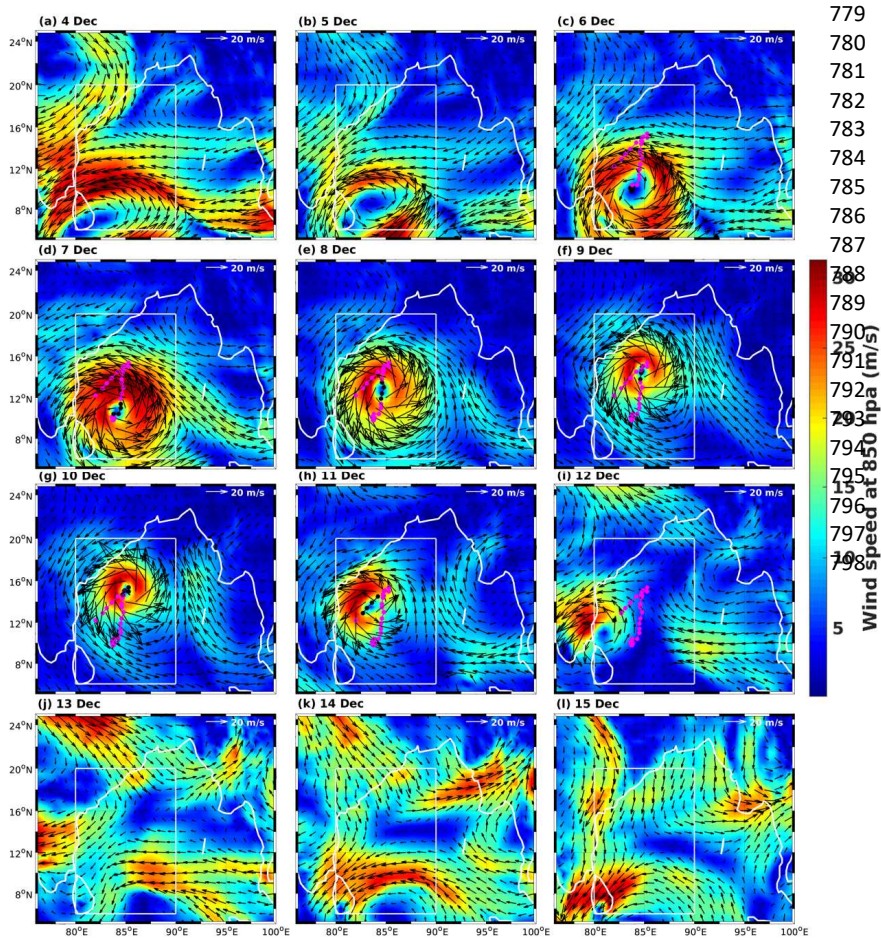


**Figure 4** Spatial maps of wind speed (shading, m/s) overlaid with wind vectors (thin arrow) at 850 hpa from 4ʰ (a) to 15ʰ (l) December 2013 with track of the cyclone overlaid. The black filled circles represent the position of the cyclone on a particular day, while the magenta filled circles indicate the track.





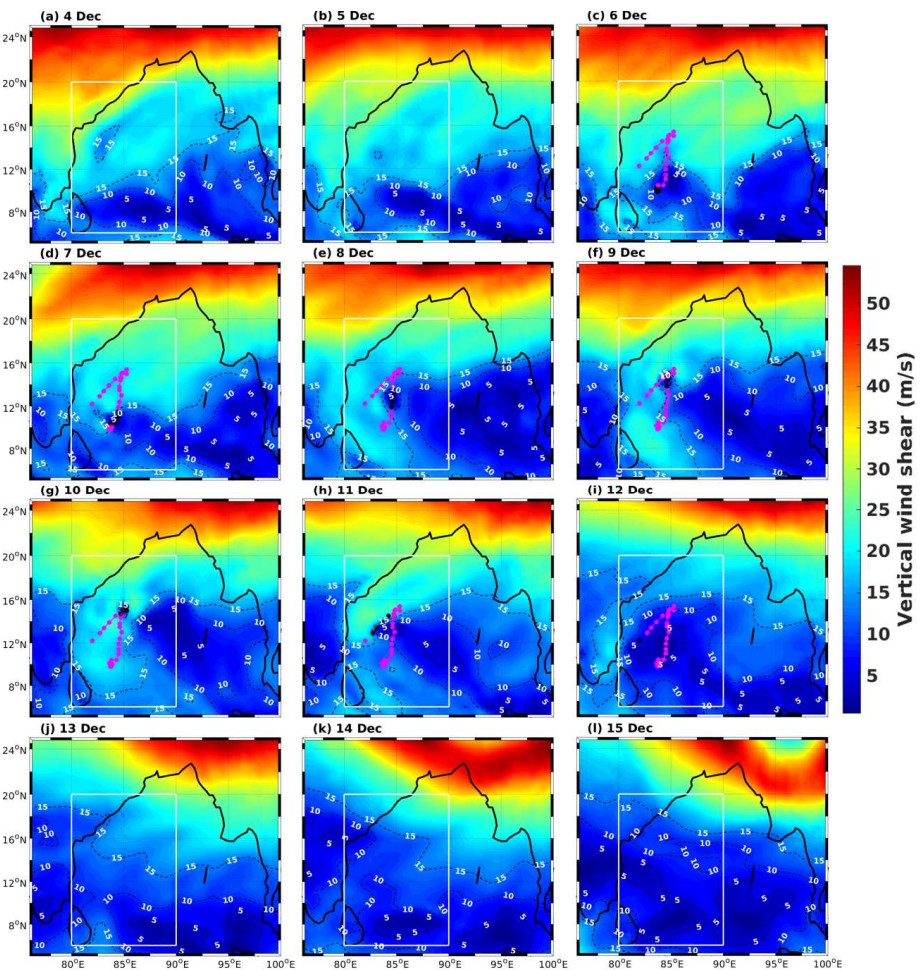

**Figure 5** Spatial maps of vertical wind velocity difference between the 850 and 200hPa
(shading, m/s) from 4th (a) to 15th (l) December 2013 with track of the cyclone overlaid.
The black filled circles represent the position of the cyclone on a particular day, while the
magenta filled circles indicate the track.



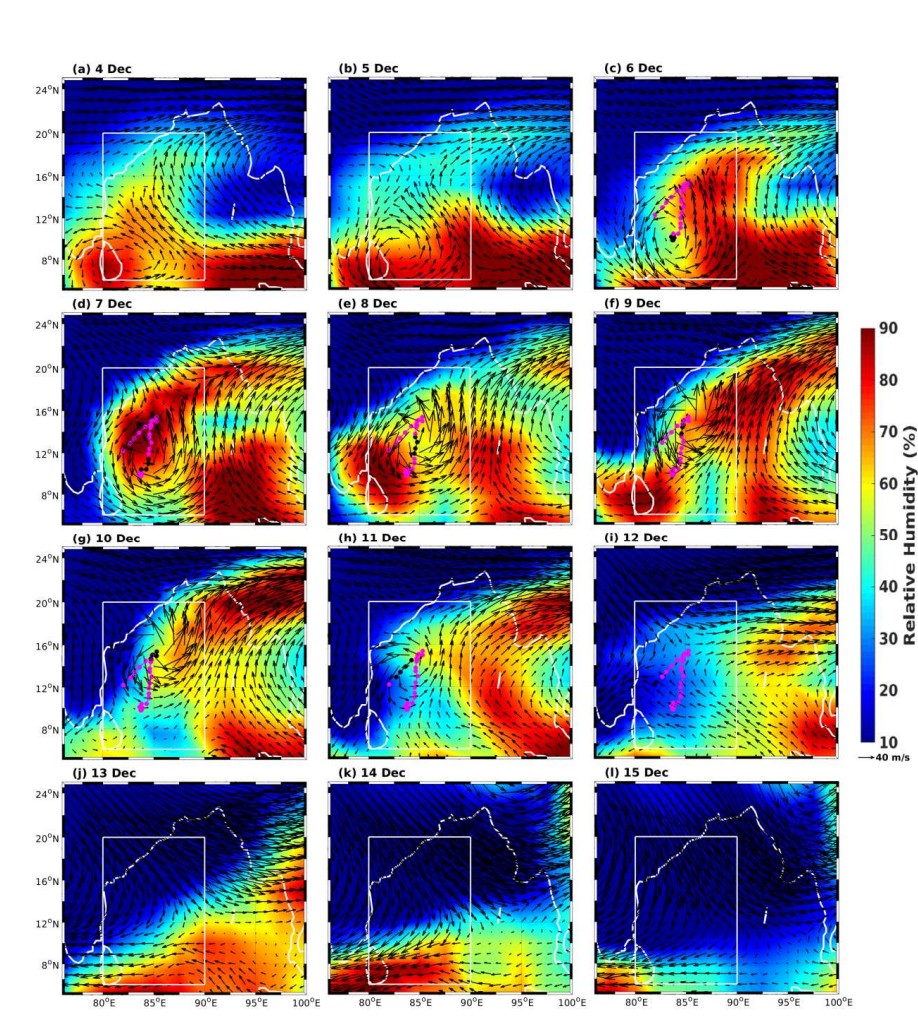


**Figure 6** Spatial maps of relative humidity (%) overlaid with winds at mid-troposhere (500 hpa) from 4[th] (a) to 15[th] (l) December 2013 with track of the cyclone overlaid. The black filled circles represent the position of the cyclone on a particular day, while the magenta filled circles indicate the track.






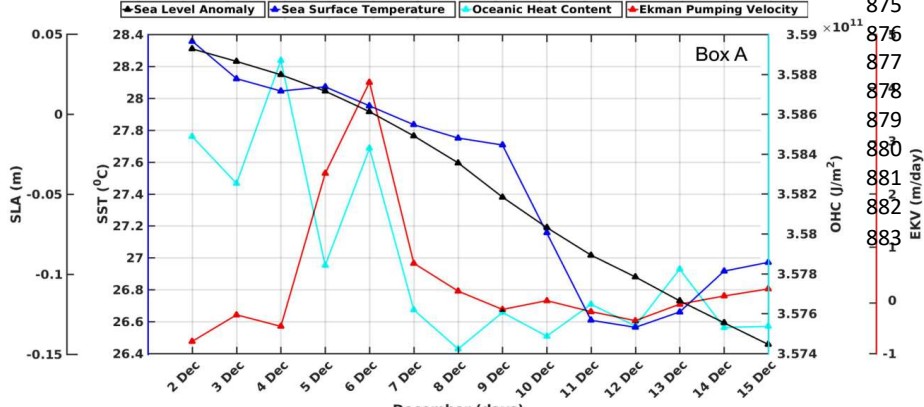

**Figure 7** Space-averaged variation of the sea surface temperature (SST, $^{o}$C), Ekman pumping velocity (EKV, m/day, positive upward), oceanic heat content (OHC, x $10^{11}$ J/m$^2$) and sea level anomaly (SLA, m) in Box A from 2-15 December 2013.


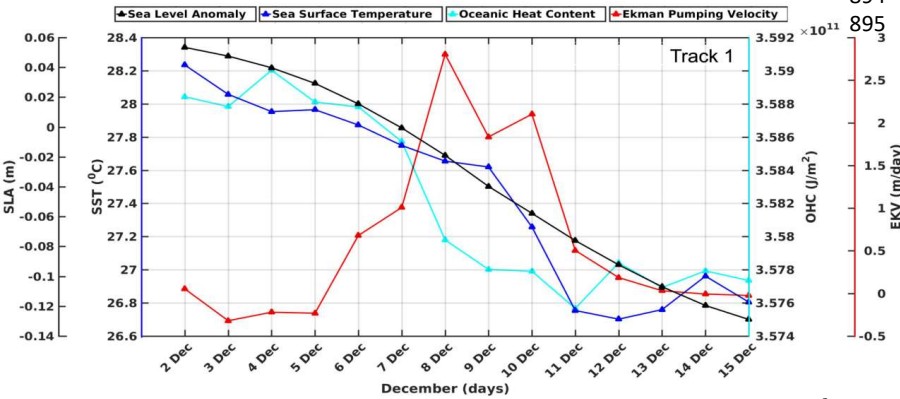

**Figure 8** Along track variation of  the sea surface temperature (SST, $^{o}$C), Ekman pumping velocity (EKV, m/day, positive upward), oceanic heat content (OHC, x $10^{11}$ J/m$^2$) and sea level anomaly (SLA, m) along Track 1 from 2-15 December 2013. These are daily averages along the track.




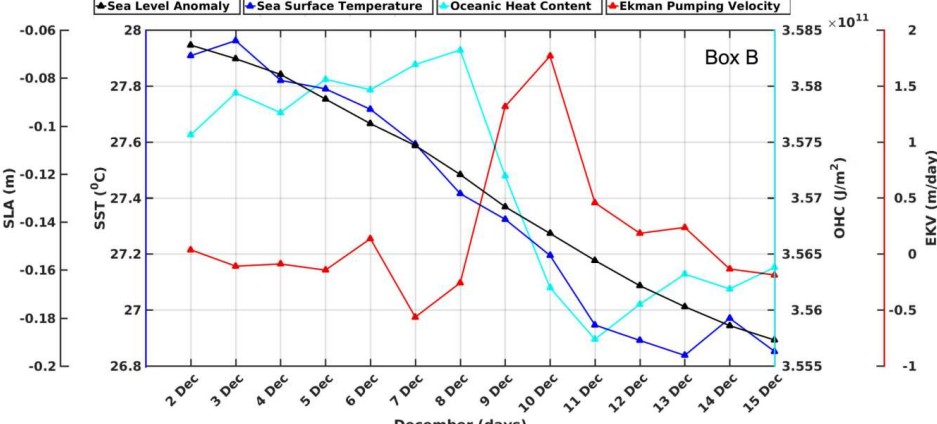


**Figure 9** Space-averaged variation of the sea surface temperature (SST, $^{o}$C), Ekman pumping velocity (EKV, m/day, positive upward), oceanic heat content (OHC, x $10^{11}$ J/m$^{2}$) and sea level anomaly (SLA, m) in Box B from 2-15 December 2013.


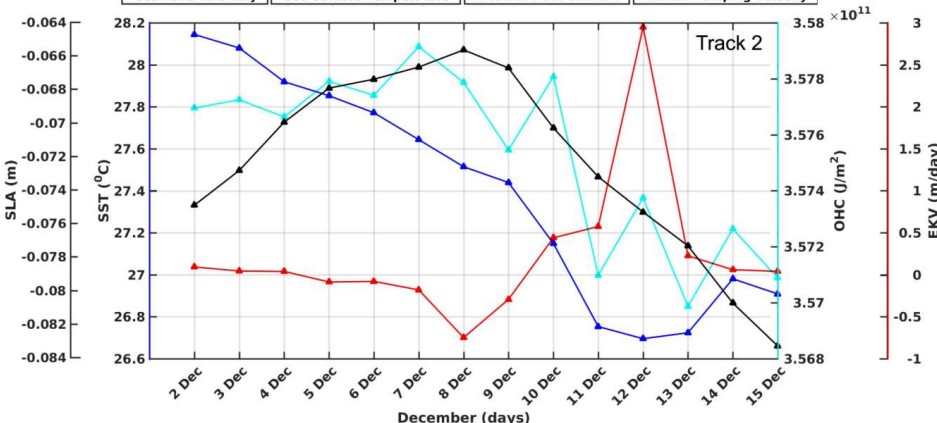


**Figure 10** Along track variation of the sea surface temperature (SST, $^{o}$C), Ekman pumping velocity (EKV, m/day, positive upward), oceanic heat content (OHC, x $10^{11}$ J/m$^{2}$) and sea level anomaly (SLA, m) along Track 2 from 2-15 December 2013. These are daily averages along the track.



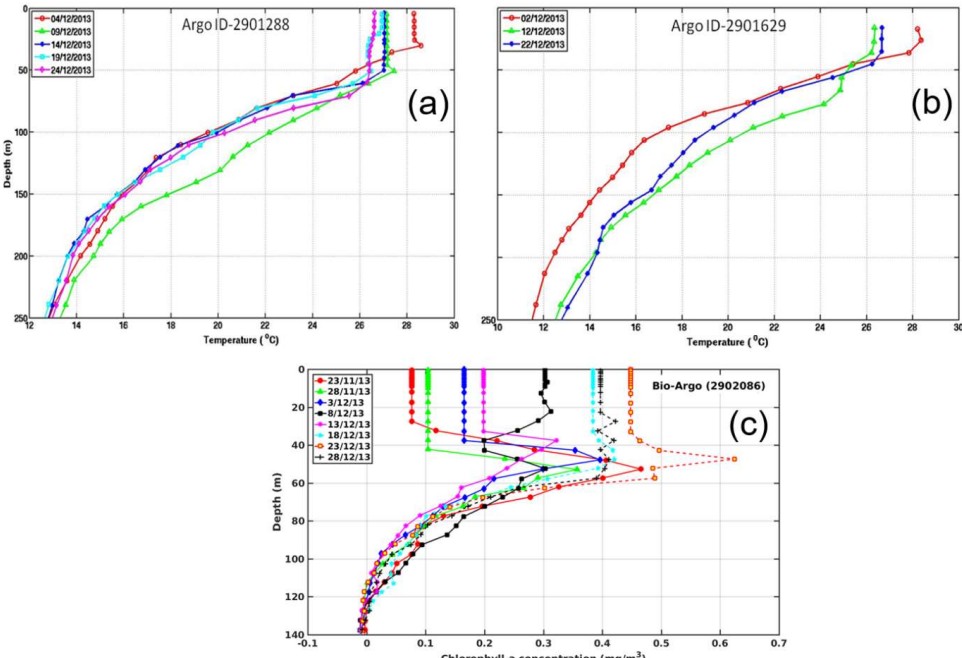

**Figure 11** Time-series of the vertical profiles of temperature ($^{o}$C) in the vicinity of Track
2 obtained from (a) Argo float ID-2901288 for 4, 9, 14, 19 and 24 December 2013, (b)
Argo float ID-2901629 for 2, 12 and 22 December 2013 and (c) chlorophyll $a$ (mg/m$^{3}$) in
the vicinity of Track 1 obtained from Bio-Argo ID-2902086 for 23 and 28 November and
3, 8, 13, 18, 23 and 28 December 2013.


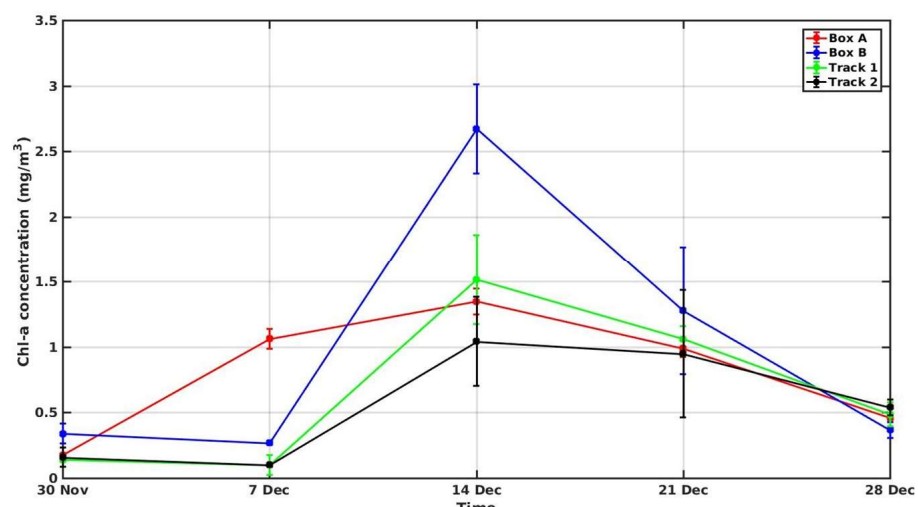

**Figure 12** Time variation of weekly composite of chlorophyll *a* pigment concentrations
(Chl-*a*, mg/m$^3$) in the Box A (red) and B (blue) and along Track 1 (green) and 2 (black)
from 30 November to 28 December 2013. The vertical lines are the standard deviations.


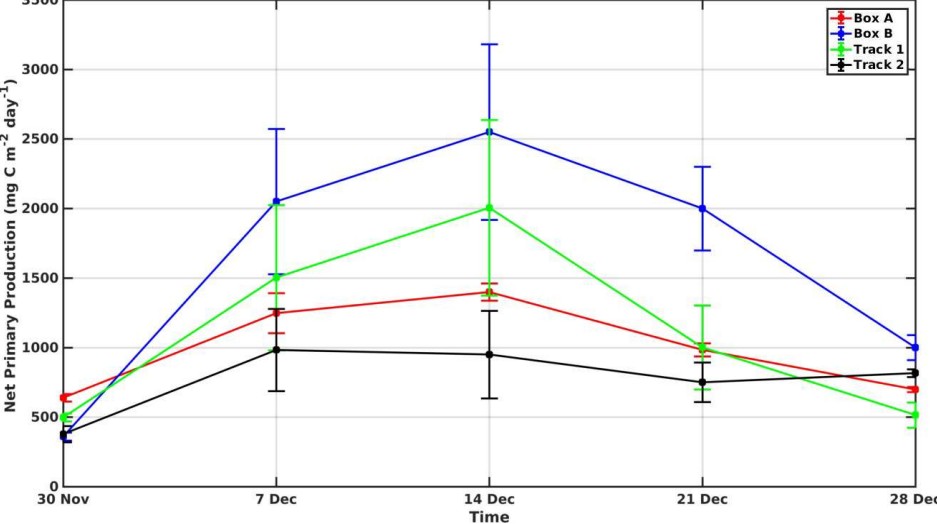


**Figure 13** Time variation of weekly composite of net primary production (NPP, mg C m$^{-2}$
day$^{-1}$) in the Box A (red) and B (blue) and along Track 1 (green) and 2 (black) from 30
November to 28 December 2013. The vertical lines are the standard deviations.







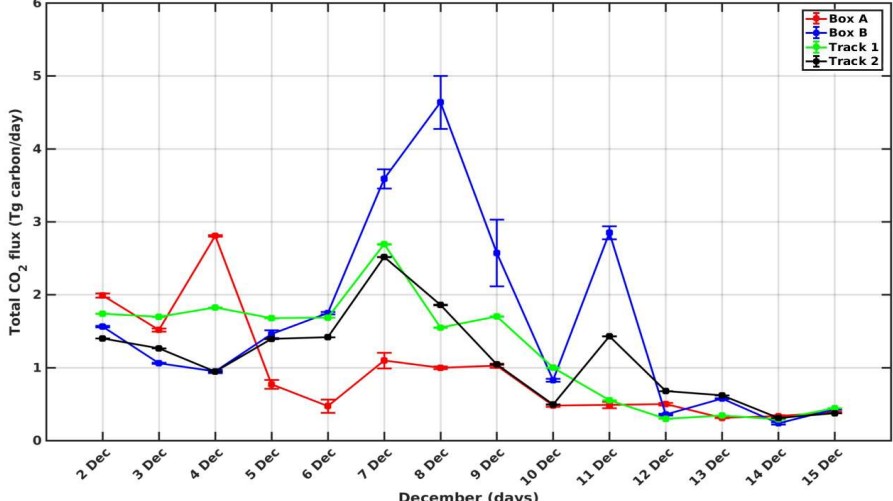

**Figure 14** Daily variation total $CO_2$ flux (terra gram carbon per day) in the Box A (red)
and B (blue) and along Track 1 (green) and 2 (black) from 2 to 15 December 2013. The
vertical lines are the standard deviations.









