# Peer review of "Eddy-induced Track Reversal and Upper Ocean Physical Biogeochemical Response of Tropical Cyclone Madi in the Bay of Bengal"

_Ocean Science, 2018_

## Referee Comment (RC1) · Anonymous Referee #1 · 31 Jan 2019

Recommendation: Substantial revision

General comments

This paper studies the upper ocean physical and biogeochemical response to Cyclone Madi (2013) in the Bay of Bengal from the analysis of multiple atmospheric and oceanic data. The topic is interesting. My understanding is that there are many studies regarding the topic: the upper ocean physical and biogeochemical response to cyclones. Some of the studies have been referenced in this manuscript. Therefore, my concern

is the novelty of this paper.

There are a few discussion points on this study.

1) I don't agree with the conclusion that "oceanic eddies affect the translation speed of Madi". This study lacks discussions on what factors do affect the movement of cyclones. In addition, there is no information regarding the atmospheric steering flow and wave-1 asymmetry of the inner core structure that is very important in understanding the movement of storms, although the authors showed the horizontal distributions of 500-hPa winds in Fig. 6. There is a cause-result confusion on the authors' conclusion that the motion of the cyclone is mainly controlled by cold eddy when the cyclone stagnated over the cold eddy without any considerations of the variation of environmental steering flow.

Looking at Fig.5, there seems to be no interaction between Madi and mid-latitude westerlies. I don't understand the role of vertical wind shear shown in Fig. 5 in the evolution of Madi. According to Fig. 6, the movement of Madi seems to be affected by high-pressure area east of the cycle, moisture transport along the edge of the high-pressure area, and high-pressure area west of the cyclone with dry airs after the recurvature (north-easterlies pointed out by Bhattacharya et al. 2015). In my opinion, the synoptic variations did affect the movement of Madi rather than the cold eddy underneath the cyclone. In that sense, I agree with Bhattacharya et al. (2015).

On the other hand, the impact of cold eddies on cyclone intensity is well known. What are new findings regarding TC-ocean interactions in this study?

2) I do not understand how to calculate net primary production (NPP). Anyway, the study of the upper ocean biogeochemical response to Cyclone Madi is one of the topics of this study. As for the sudden increases in total $CO_2$ flux, Bates et al. (1998) and Nemoto et al. (2009) have already reported it from the analysis based on the observations. In addition, Wada et al. (2011) conducted numerical simulations by an ocean general circulation model to clarify the mechanism and relation to the relative

position to the storm center regarding the variation of the sudden increases in total $CO_2$ flux. My question is what are new findings in this study regarding the relation of oceanic eddies to the sudden increases in total $CO_2$ flux. This also applies to chlorophyll-a concentrations. In the abstract, the authors only show the number of folds regarding rapid increases in chlorophyll-a concentrations, NPP and total $CO_2$ flux. Without any information on these background values, readers cannot understand the importance of these extreme events.

In that sense, the authors did not present in-depth analysis on the time series of the vertical profiles of water temperature, salinity and chlorophyll-a concentrations. These profiles may show the ocean response to a storm occurred on the right-hand side of a moving storm (in the Northern Hemisphere). Certainly, sea surface temperature in the mixed layer decreased due to vertical turbulent mixing, but sea surface temperature in the thermocline increased during the passage of the storm and then decreases. There is no discussion regarding this oceanic response and its relation to variation in chlorophyll-a concentrations although they have been already studied.

I hope the above mentioned discussion points will be clear.

Specific comments

3) I suspect the calculation method (equation (3)) of ocean heat content (OHC) in this study. First, there is no information on the oceanic dataset in the manuscript. At least, water temperature from the surface to 300-m depth is needed in equation (3). Second, 3.574x1011 (Q) / 1.026x103 (density) / 3.87x103 (Cp) / 300(m) $\sim$ 300 K. This means that water temperature should be 300K from the surface to 300-m depth so that this result is not consistent with Fig. 11(a). Therefore, Fig. 3 only shows information which is the same as the horizontal distribution of sea surface temperature.

4) In equations (1) and (2), there is no information regarding how to calculate wind stresses.

[Figure]

5) In the abstract, there are many descriptions regarding a lifecycle of Madi. It seems that the authors intentionally increased the number of characters of the abstract.

6) The reference of O'Brian et al. is not correct. Maybe this paper shows the results of idealized numerical experiments regarding the upper ocean response to a vortex.

7) In general, vertical wind shear is calculated as an areal average. How did the authors calculate the vertical wind shear? What did the authors intend to analyze the vertical wind shear?

8) How did the authors calculate weekly composite of daily chlorophyll-a concentrations? Because the moving speed of Madi is relatively slow, there seem to be a lot of data missing areas.

9) In equation (5), salinity data are required. However, there is no information on salinity data used in this study.

10) In Table 1, the category of the intensity of Madi is not based on the Saffir-Simpson scale.

11) It is very hard to see black arrows in Figs. 3-6. The marks in all figures tend to be small.

References (* new or revised reference) Bates, N. R., Knap, A. H., Michaels, A. F.: Contribution of hurricanes to local and global estimates of air-sea exchange of $CO_2$, Nature, 395, 58-61, 1998.

Bhattacharya, S. K., Kotal, S. D., Kundu, P. K.: An Analysis of recurvature and decay of the tropical cyclone 'MADI' over the Bay of Bengal, Tropical Cyclone Research and Review, 4, 27-37, https://doi.org/10.6057/2015tcRR01.04, 2015.

*Nemoto, K., Midorikawa, T., Wada, A., Ogawa, A., Takatani, S., Kimoto, H., Ishii, M., Inoue, H.Y.: Continuous observations of atmospheric and oceanic $CO_2$ using a moored buoy in the East China Sea: Variations during the passage of typhoons, Deep-Sea

Research II, 56, 542-553, 2009.

*Wada, A., Midorikawa, T., Ishii M., and Motoi, T.: Carbon system changes in the East China Sea induced by Typhoons Tina and Winnie in 1997. Journal of Geophysical Research - Oceans, 116, C07014, 2011.

---

## Short Comment (SC1) · 8 Mar 2019

**Reply to the Comments of Anonymous Referee #1**

We thank the Anonymous Referee#1for reviewing our manuscript and for suggesting improvements. We have addressed all the concerns of the Referee#1.

**General comments**

**• Referee's Comment**

This paper studies the upper ocean physical and biogeochemical response to Cyclone Madi (2013) in the Bay of Bengal from the analysis of multiple atmospheric and oceanicdata. The topic is interesting. My understanding is that there are many studies regarding the topic: the upper ocean physical and biogeochemical response to cyclones. Some of the studies have been referenced in this manuscript. Therefore, my concern is the novelty of this paper.

**Author's Response**

In this paper we not only bring out the cyclone-induced physical and biogeochemical response of the surface ocean using a suite of remote sensing data, we document the timeevolving depth-dependent temperature and chlorophyll response of the upper ocean to tropical cyclone, which is the novelty of this paper. We have used two Argo float data (WMO ID 2901288, 2901629) for temperature profiles that were in the vicinity of Track 2 and one Bio-Argo float (WMO ID 2902086) for chlorophyll profiles that was to the right of Track 1. Another novelty is that the enhancement of Chl-*a* and net primary productivity by cyclone Madi is much greater compared with previous cyclones that occurred in the BoB. We show that this is the combined impact of cyclone and oceanic cyclonic eddy. This is significant because BoB is known for its low upper ocean Chl-*a* and primary productivity.

**• Referee's Comment**

There are a few discussion points on this study.

1) I don't agree with the conclusion that "oceanic eddies affect the translation speed of Madi". This study lacks discussions on what factors do affect the movement of cyclones.In addition, there is no information regarding the atmospheric steering flow andwave-1 asymmetry of the inner core structure that is very important in understanding the movement of storms, although the authors showed the horizontal distributions of 500-hPa winds in Fig. 6. There is a cause-result confusion on the authors' conclusionthat the motion of the cyclone is mainly controlled by cold eddy when the cyclone stagnated over the cold eddy without any considerations of the variation of environmentalsteering flow.

**Author's Response**

To address the concern of the reviewer, especially the role of atmospheric steering, we wish to clarify this point with the help of additional computations and figures. We could include a summary of the following description, in very brief, in the discussion part of the modified manuscript.

Environmental conditions that are necessary for the formation of tropical cyclones are (1) SST in excess of  $26.5^{\circ}$ C, (2) low level wind convergence, (3) high relative vorticity, (4) weak vertical wind shear, and (5) high relative humidity.

Once formed cyclones generally move in a westerly/north-westerly direction due to the advection of potential vorticity field by storm circulation. In simple case, the background potential vorticity gradient is simply the meridional gradient of the Coriolis parameter (f) where  $f = 2\Omega \sin \Phi$  and  $\Phi$  varies with the latitude, beta  $(\beta = \frac{\partial f}{\partial y})$ . The Beta drift generally causes the tropical cyclone to move poleward and westward relative to the motion they would have if the background potential vorticity field were unperturbed by the storms.

Based on the comments by the reviewer, in order to examine the role of environmental steering flow, we show the analysis of the vorticity dynamics, which is the tendency of the absolute vorticity term  $(\frac{\partial \eta}{\partial t})$ , as per the following equation:

$$\frac{\partial \eta}{\partial t} = -\vec{v} \cdot \vec{\nabla} \left(\zeta + f\right) - \left(\zeta + f\right) \left(\nabla \cdot \vec{v}\right) - w \frac{\partial \varepsilon}{\partial p} + \left(\frac{\partial w}{\partial x} \frac{\partial v}{\partial z} - \frac{\partial w}{\partial y} \frac{\partial u}{\partial z}\right) + \vec{F}$$
(1)
(2)
(3)
(4)
(5)

Where, u,v,w are the velocity component. The vertical component  $\eta$  of the absolute vorticity vector (as defined above) given by the sum of the vertical component of vorticity with respect to the earth (relative vorticity ( $\zeta$ )) and the vorticity of the earth (equal to the Coriolis parameter *f*).

$$\zeta = \frac{\partial v}{\partial x} - \frac{\partial u}{\partial y}$$
 where u and v are zonal and meridional wind.
 $f = 2\Omega \sin \Phi$  where  $\Phi$  varies with the latitude, and  $\eta = \zeta + f$

The local change of absolute vorticity is related to horizontal advection (term 1), stretching term (term 2; divergence multiplied by absolute vorticity), vertical advection (term 3), tilting term (term 4)and friction (term 5). The contribution from the each term in the absolute vorticity tendency equation was examined. The analysis has shown that at lower level (850hpa) stretching and horizontal advection in the vorticity equation are the major contributors of the absolute vorticity tendency term. The contribution from other terms being very small could be neglected.

Low level vorticity being one of the environmental factor which plays a key role in the formation and development of the tropical cyclone, the time evolution of vorticity tendency (Fig.A), horizontal advection (Fig.B) and stretching term (Fig.C), all at 850 hpa, were analyzed. Note that the absolute vorticity tendency in the area where the genesis of cyclone Madi took place was high before the genesis Madi and indicated the prevailing atmospheric condition congenial for its formation. The track of the cyclone moves through the region of high absolute vorticity tendency. Consistent with this the horizontal advection and the stretching term also showed higher values in the vicinity of the genesis of cyclone. The stretching term is related to absolute vorticity and divergence of wind. Increased divergence is related to increased convection and generation of more low pressure which is favourable

for cyclone. On the 9-10th December when the northward movement of cyclone Madi was arrested the region showed increased absolute vorticity tendency and horizontal advection with increase in stretching term, all of which supported the strengthening and northward progression of Madi. However, contrary to this, the cyclone slowed down and finally became stationary. Recall that at this time the cyclone Madi was passing through the region of cold-core eddy.

---

## Referee Comment (RC2) · Anonymous Referee #1 · 11 Mar 2019

I am grateful for the authors' replies. Although the presentation of the paper might have been improved, I can not confirm it at the present moment, because I cannot find the revised paper in the web now. Therefore, my comments are only for the replies.

For the biogeochemical oceanic response to a storm, I understand that the observation itself is new. However, the authors did not reply (and revise) the following scientific themes: What kind of processes did the oceanic response occur by? What effects did the response have on?

[Figure]

I also understand that there was a sudden change in the biogeochemical components such as Chl-a and outgassing of $CO_2$ from the background before the passage. However, the authors have not shown any evidence of the difference quantatively from the present study to Bate et al. (1998) and Nemoto et al. (2009). According to Wada et al. (2011), the amount of outgassing of $CO_2$ is greatly affected by the error of surface wind speed analysys data. Therefore, I think that the difference the authors found is not a new finding but the result including the observational error.

I am not convinced that the cold eddy controlled the movement of the typhoon (arresting of the northward movement). I understand that the results of absolute vorticity budget analysis do show the processes that dominates the storm by solving the given atmospheric field diagnostically. However, I confirm that the authors' analysis did not clarify the relation between the vorticity balance and oceanic cold eddy scientifically although the timing that a storm was arresting of the northward movement matched when a storm was over the cold eddy.

The fact that the vertical shear is small and the moving speed is slow is also related to the environmental steering flow of the storm. In such an environmental field, the axis of the storm tends to stand up, the storm weakened due to sea surface cooling, and the influence on the inner-core structure become more clear. This is a well known mechanism about tropical cyclone-ocean interactons. However, the authors do not show the influence of the intensity and structure of the storm on the environmental steering flow. Therefore, the authors do not demonstrate the mechanism regarding the arresting of the northward movement over the cold eddy. Conversely, it is easy to understand the arresting of the northward movement over the cold eddy led to decreases in TCHP and increases in Chl-a and outgassing of $CO_2$.

Therefore, I recommend that this paper in the present form is not worthy of publication.

---

## Short Comment (SC2) · 25 Mar 2019

Reply to the Comments of Anonymous Referee #1 posted on 11 March 2019

General Comment

âǍć Referee's General Comment

I am grateful for the authors' replies. Although the presentation of the paper might havebeen improved, I can not confirm it at the present moment, because I cannot find

therevised paper in the web now. Therefore, my comments are only for the replies.

Author's Response

We have gone through the comments of the Referee#1 very carefully. In fact, there are 4 comments this time and we addressed them point-by-point in the following section. However, all the 4 points broadly converge to Referee's reviews about (1) role of cold core eddy in controlling/arresting the northward movement of cyclone Madi and (2) how the present study on the out-gassing of CO2 is quantitatively different from that of earlier studies.

(1) Role of cold core eddy in controlling/arresting the northward movement of cyclone Madi It is unfortunate that Referee#1 still remains unconvinced about the role of cold core eddy in controlling the northward movement of the cyclone.

To quantify the eddy's contribution to the intensity of the cyclone Madi, we have calculated the eddy feedback factor following Wu et al. (2007). The analysis showed (for details see following section on Reply to Reviewer's specific comments) that from 7th to 8th December when the system intensified from CS to VSCS and was passing through the warm patch associated with warm core eddy (see spatial maps of OHC at Fig.2 & positive SLA at Fig.3 of un corrected manuscript) the eddy feedback factor was positive and amounted to 59%. Thereafter, when the cyclone passed over the cold patch associated with cold core eddy during 9thand 10th December, the eddy feedback factor was negative and 69%. Thus, this analysis quantifies the contribution of both warm and cold core eddies; when cyclone passed through the warm patch the system intensified from CS to VSCS and its translation speed increased (see Table 1), while when it passed over the cold patch from 9th to 10th the system slowed down and its northward movement was arrested as noted under the section 2.3. We have elaborated the methodology of computation of eddy feedback factor under the "Response to specific comments" with a new diagram. The modification to the manuscript is also indicated there. We hope this will convince the Reviewer#1

2) How the present study on the out-gassing of CO2 is quantitatively different from that of earlier studies Based on the Reviewer's comment, in order to compare our CO2 flux with that of previous studies, we have recomputed the CO2 flux along Track 1, Track 2 and Boxes A and B in mmolper meter square per day. While re-computing CO2 flux we noticed a bug in our previous calculation, which we rectified. The newly calculated values showed a cyclone-induced CO2 out-gassing which was about 4-times greater than the pre-cyclone values along Track 1 and in Box B. The impact of CO2 out-gassing in Box A and along Track 2 were much smaller as when the cyclone was in this box and was passing through this track it was in a formative and dissipative stage respectively.

Several studies have demonstrated that the passage of a tropical cyclone can lead to enormous amount of CO2 flux from the ocean surface to the atmosphere. For example, based on observation from Sargasso Sea during summer 1995 Bates et al. (1998) showed that hurricanes accounted for nearly 55% of the CO2 flux into the atmosphere, while based on moored buoy data from the East China Sea Nemoto et al. (2009) reported a 60% contribution from typhoon in summer. In the eastern Arabian Sea Byju and Prasanna Kumar (2011) noted that cyclone Phyan emitted $\sim$8 mmol m2 day1 of CO2 from ocean to atmosphere accounting for $\sim$85% of the total out-gassing for the month of November (climatology) calculated by Takahashi et al. (2009). Our study show that during cyclone Madi (6-12 Dec) Maximum CO2 flux observed was 13 mmol m-2 day-1. Tropical cyclones have significant impact on the carbon cycle in the Bay of Bengal (Ye et al., 2019). Based on their study cyclone Hudhud and cyclone Roanu formed over the Bay of Bengal enhanced CO2 efflux (18.49 $\pm$ 3.70 mmol CO2 m-2 day-1) and (19.08 $\pm$ 3.82 mmol CO2 m-2 day-1) due to wind effect during the storm.

We have elaborated this under Response to specific comments with a new diagram. The modification to the manuscript is also indicated there.

Reply to the Specific Comments of Reviewer#1 along with figures and Table is uploaded as Supplement pdf.

Please also note the supplement to this comment:
https://www.ocean-sci-discuss.net/os-2018-133/os-2018-133-SC2-supplement.pdf

**Supplement:**

**Reply to the Comments of Anonymous Referee #1posted on 11 March 2019**

**General Comment**

**• Referee's General Comment**

I am grateful for the authors' replies. Although the presentation of the paper might havebeen improved, I can not confirm it at the present moment, because I cannot find therevised paper in the web now. Therefore, my comments are only for the replies.

**Author's Response**

We have gone through the comments of the Referee#1 very carefully. In fact, there are 4 comments this time and we addressed them point-by-point in the following section. However, all the 4 points broadly converge to Referee's reviews about (1) role of cold core eddy in controlling/arresting the northward movement of cyclone Madi and (2) how the present study on the out-gassing of  $CO_2$  is quantitatively different from that of earlier studies.

**(1) Role of cold core eddy in controlling/arresting the northward movement of cyclone Madi**

It is unfortunate that Referee#1 still remains unconvinced about the role of cold core eddy in controlling the northward movement of the cyclone.

To quantify the eddy's contribution to the intensity of the cyclone Madi, we have calculated the eddy feedback factor following Wu et al. (2007). The analysis showed (for details see following section on Reply to Reviewer's specific comments) that from 7th to 8th December when the system intensified from CS to VSCS and was passing through the warm patch associated with warm core eddy (see spatial maps of OHC at Fig.2 & positive SLA at Fig.3 of un corrected manuscript) the eddy feedback factor was positive and amounted to 59%. Thereafter, when the cyclone passed over the cold patch associated with cold core eddy during 9th and 10th December, the eddy feedback factor was negative and 69%. Thus, this analysis quantifies the contribution of both warm and cold core eddies; when cyclone passed through the warm patch the system intensified from CS to VSCS and its translation speed increased (see Table 1), while when it passed over the cold patch from 9th to 10th the system slowed down and its northward movement was arrested as noted under the section 2.3. We have elaborated the methodology of computation of eddy feedback factor under the "Response to specific comments" with a new diagram. The modification to the manuscript is also indicated there. We hope this will convince the Reviewer#1

**2) How the present study on the out-gassing of $CO_2$ is quantitatively different from that of earlier studies**

Based on the Reviewer's comment, in order to compare our  $CO_2$  flux with that of previous studies, we have recomputed the  $CO_2$  flux along Track 1, Track 2 and Boxes A and B in

mmolper meter square per day. While re-computing  $CO_2$  flux we noticed a bug in our previous calculation, which we rectified. The newly calculated values showed a cyclone-induced  $CO_2$  out-gassing which was about 4-times greater than the pre-cyclone values along Track 1 and in Box B. The impact of  $CO_2$  out-gassing in Box A and along Track 2 were much smaller as when the cyclone was in this box and was passing through this track it was in a formative and dissipative stage respectively.

Several studies have demonstrated that the passage of a tropical cyclone can lead to enormous amount of CO2 flux from the ocean surface to the atmosphere. For example, based on observation from Sargasso Sea during summer 1995 Bates et al. (1998) showed that hurricanes accounted for nearly 55% of the CO2 flux into the atmosphere, while based on moored buoy data from the East China Sea Nemoto et al. (2009) reported a 60% contribution from typhoon in summer. In the eastern Arabian Sea Byju and Prasanna Kumar (2011) noted that cyclone Phyan emitted ~8 mmol m2 day1 of CO2 from ocean to atmosphere accounting for ~85% of the total out-gassing for the month of November (climatology) calculated by Takahashi et al. (2009). Our study show that during cyclone Madi (6-12 Dec) Maximum CO2 flux observed was 13 mmol m-2 day-1. Tropical cyclones have significant impact on the carbon cycle in the Bay of Bengal (Ye et al., 2019). Based on their study cyclone Hudhud and cyclone Roanu formed over the Bay of Bengal enhanced CO2 efflux (18.49 ± 3.70 mmol  $CO_2 m^{-2} day^{-1}$ ) and (19.08 ± 3.82 mmol  $CO_2 m^{-2} day^{-1}$ ) due to wind effect during the storm.

We have elaborated this under Response to specific comments with a new diagram. The modification to the manuscript is also indicated there.

**Specific Comments and Point-wise Reply to Referee#1**

**Author's Response**

**• Referee's Comment**

For the biogeochemical oceanic response to a storm, I understand that the observationitself is new. However, the authors did not reply (and revise) the following scientificthemes: What kind of processes did the oceanic response occur by? What effects didthe response have on?.

**Author's Response**

Sorry to say that we did not quite understand the question of the Reviewer. Assuming that the Reviewer is enquiring about the oceanic processes that are responsible for the response in terms of enhanced chlorophyll concentration and enhanced  $CO_2$  out-gassing, following is our reply.

As the tropical cyclone Madi passes over the BoB, the upper ocean experiences strong vertical mixing associated with strong winds, which is essentially a one-dimensional response. In addition to this, the cyclonic winds lead to strong Ekman divergence, which is a three-dimensional response. This, in turn, forcesthe subsurface cold and nutrient rich waters to come to the surface under the upward Ekman pumping. Increased availability of nutrients to the upper ocean will initiate the carbon fixation by phytoplankton in the euphotic zone and results in the enhancement in chlorophyll biomass. In addition to the enhancement of chlorophyll biomass with time, the upward Ekman pumping also would result in an increase in the CO2 out-gassing in the following manner. As the cold subsurface waters comes to the surface it also brings with it higher concentration of dissolved CO2. Once at the surface, the warmer temperature and strong winds will initiate a strong out-gassing of CO2 from ocean surface to the atmosphere. This happens under the action of all tropical cyclone. What is distinct in our case is that the cyclone Madi is passing over cold core eddy. Under this condition the upward transport of  $CO_2$  rich subsurface water occurs due to both Ekman pumping driven by cyclonic winds associated with the cyclone Madias well eddy-pumping driven by cyclonic circulation of water in a cold core eddy. Accordingly, in our study we see a 4-fold increase in the CO2 out-gassing compare to its pre-cyclone values, when the cyclone passes over track 1 and Box B which has cyclonic eddy. See more details under the next reply.

**• Referee's Comment**

I also understand that there was a sudden change in the biogeochemical componentssuch as Chl-a and outgassing of CO2 from the background before the passage. However, the authors have not shown any evidence of the difference quantitatively from the present study to Bate et al. (1998) and Nemoto et al. (2009). According to Wada etal. (2011), the amount of outgassing of CO2 is greatly affected by the error of surfacewind speed analysis data. Therefore, I think that the difference the authors found isnot a new finding but the result including the observational error.

**Author's Response**

The air-sea  $CO_2$  flux at the sea surface depend on the difference between the partial pressure of  $CO_2$  at the sea surface (p $CO_2^{sea}$ ) and in the overlying atmosphere (p $CO_2^{air}$ ), the wind speed, sea surface temperature and sea surface salinity as per the equations 4, 5 and 6 in the manuscript. Among these factors, the wind speed plays an important role in determining the value of air-sea  $CO_2$  flux due to the quadratic functional dependence of the gas transfer velocity with the wind speed. Several studies have demonstrated that the passage of a tropical cyclone can lead to enormous amount of  $CO_2$  flux from the ocean surface to the atmosphere. For example, based on observation from Sargasso Sea during summer 1995 Bates et al. (1998) showed that hurricanes accounted for nearly 55% of the  $CO_2$  flux into the atmosphere, while based on moored buoy data from the East China Sea Nemoto et al. (2009) reported a 60% contribution from typhoon in summer. In the eastern Arabian Sea Byju and Prasanna Kumar (2011) noted that cyclone Phyan emitted ~8 mmol m2 day1 of CO2 from ocean to atmosphere accounting for ~85% of the total out-gassing for the month of November (climatology) calculated by Takahashi et al. (2009). Our study show that during cyclone Madi (6-12 Dec) maximum CO2 flux observed was 13 mmol m-2 day-1. Tropical cyclones have significant impact on the carbon cycle in the Bay of Bengal (Ye et al., 2019). Based on their study cyclone Hudhud and cyclone Roanu formed over the Bay of Bengal enhanced CO2 efflux (18.49 ± 3.70 mmol CO2 m-2 day-1) and (19.08 ± 3.82 mmol CO2 m-2 day-1) due to wind effect during the storm.

Based on the Reviewer's comment in order to compare our  $CO_2$  flux with that of previous studies we have recomputed the  $CO_2$  flux along Track 1, Track 2 and Boxes A and B in mmol per meter square per day. While re-computing  $CO_2$  flux we noticed a bug in our previous calculation, which we rectified and a new figure is generated as given below.

Fig A. Daily variation total  $CO_2$  flux (mmol/m2/day) in the Box A (red) and B (blue) and along Track 1 (green) and 2 (black) from 2 to 15 December 2013. The vertical lines are the standard deviations.

The newly calculated values showed a cyclone-induced  $CO_2$  out-gassing which was about 4times greater than the pre-cyclone valuesalong Track 1 and Box B, i.e., from 3 to 13mmol m-2 day-1. The impact of  $CO_2$  out-gassing in Box A and along Track 2 were much smaller because when the cyclone was inBox A and passing through Track 2 it was in a formative and dissipative stages respectively.

Regarding the observational error associated with wind data used in our present study, the bias and root-mean-square differences of the wind speed between ASCAT and dropwindsonde data are -1.7 and  $5.3 \text{ ms}^{-1}$  (Chou et al., 2013). ASCAT winds are most reliable when the wind speeds are in the range of 12 and 18 m s-1 and can be applied to determine the radius of 34 knot winds, a critical parameter in operational tropical cyclone analysis (Chou et al., 2013).

In our case the wind speed used for the computation of  $CO_2$  flux ranged from 2.72 to 10.38 ms-1. We have calculated the correlation between Rama buoy and ASCAT wind data in BoB region. We have chosen wind speed data from the RAMA buoy located at 12 N 90 E and 15 N 90 E for the comparison with ASCAT wind product which is used for the CO2 flux calculation in our study. The location of the Rama buoy was nearby the track of the cyclone Global tropical Madi. Rama data was taken from moored buoy array (https://www.pmel.noaa.gov/tao/drupal/disdel/). The correlation Coefficient values are 0.89 and 0.83 respectively indicating the quality of the ASCAT data

Fig B. Comparison between wind data from RAMA buoy and ASCAT wind data for December 2013

In the modified manuscript we will include the above matter in place of the matter between lines 320 to 333 of the pre-modified version of the manuscript. We will also replace figure 13 with Fig.A.

**• Referee's Comment**

I am not convinced that the cold eddy controlled the movement of the typhoon (arrestingof the northward movement). I understand that the results of absolute vorticitybudget analysis do show the processes that dominates the storm by solving the givenatmospheric field diagnostically. However, I confirm that the authors' analysis did notclarify the relation between the vorticity balance and oceanic cold eddy scientificallyalthough the timing that a storm was arresting of the northward movement matchedwhen a storm was over the cold eddy.

The fact that the vertical shear is small and the moving speed is slow is also related to the environmental steering flow of the storm. In such an environmental field, the axis of the storm tends to stand up, the storm weakened due to sea surface cooling, and theinfluence on the inner-core structure become more clear. This is a well known mechanism about tropical cyclone-ocean interactons. However, the authors do not show theinfluence of the intensity and structure of the storm on the environmental steering flow. Therefore, the authors do not demonstrate the mechanism regarding the arresting of the northward movement over the cold eddy. Conversely, it is easy to understand thearresting of the northward movement over the cold eddy led to decreases in TCHP and increases in Chl-a and outgassing of CO2.

**Author's Response**

To quantify the eddy's contribution to the intensity of the cyclone Madi we have calculated the eddy feedback factor  $F_{EDDY-}$  following Wu et al., (2007) based on the following equation

$$F_{EDDY-} = 0.38 (SST_{Eddy}-26^{\circ}C)^{2.08} (SST-26^{\circ}C)^{-1.88} (ML_{Eddy})^{0.98} x (ML)^{-0.97} (\eta)^{0.22} (1-RH)^{-0.74} (\Gamma)^{0.45} (U_{\rm H})^{-0.83}$$

The Table below gives the description of the parameter, its value and unit used for the computation of eddy feedback factor. The values for the SST,  $SST_{Eddy}$ , ML,  $ML_{Eddy}$ , and  $\Gamma$  were obtained from the Argo float data, while the translation speed were calculated from IMD data.

| Parameter                      | Unit              | Range     |
|--------------------------------|-------------------|-----------|
| SST-26°C                       | °C                | 2.2-2.4   |
| SST Eddy -26°C      | °C                | 1-1.2     |
| Mixed layer Depth (Standard    | m                 | 20        |
| Ocean) (ML)                    |                   |           |
| Mixed layer Depth (Eddy        | m                 | 50        |
| Ocean) (ML Eddy )   |                   |           |
| Storm size $(\eta)$            | 1                 | 1         |
| Relative Humidity (1-RH)       | 1                 | 60-90%    |
| Stratification below the Mixed | °Cm -1 | 0.06      |
| layer (Γ)                      |                   |           |
| Translation speed $(U_H)$      | ms -1  | 1.63-5.41 |

Table I. Value of the parameters, their unit and range used in the calculation of eddy feedback factor.

The eddy feedback factor could be positive or negative;  $F_{EDD} = +0.5$ , indicates an increase in the storm intensity by 50% due to the interaction with the warm ocean region, while a  $F_{EDD} = -0.5$  indicates a decrease in storm intensity by 50% due to the interaction with cold ocean region (Wu et al., 2007).

The analysis showed that from 7th to 8th December when the system intensified from CS to VSCS and was passing through the warm patch associated with warm core eddy (see spatial maps of OHC at Fig.2 & positive SLA at Fig.3) the eddy feedback factor was positive and amounted to 59%. Thereafter, when the cyclone passed over the cold patch associated with cold core eddy during 9th and 10th December, the eddy feedback factor was negative and 69%. The figure below (Fig. C) pictorially represents the time evolution of the estimated

central pressure (hpa) andmaximum sustained surface wind (in knots) of the cyclone Madi along with eddy feedback factor.

---

## Referee Comment (RC3) · Anonymous Referee #2 · 26 Mar 2019

General Comments:

Based a suite of atmospheric and oceanic datasets during the passage of TC Madi, Chowdhury et al. examined the upper ocean physical-biogeochemical response to the TC, mostly emphasized the effect of pre-existing cold core eddies underneath the TC. The topic of TC-ocean interaction in the BoB is interesting and important for TC forecasting. Generally, the effect of mesoscale eddy on TC-ocean interaction is well known at the present stage. Due the lack of in situ observations, studies on the Biogeochemical response to a TC is relatively less and this study may enrich our knowledge on the

biogeochemical change induced by TC passage.

In the manuscript, I find some conclusions are inaccurate or unclear with not sufficient evidences, especially on the effect of mesoscale eddies. Therefore, I suggest a major revision prior publication. I hope the following comments are useful when the authors revise their manuscript.

(1) How does cyclonic eddy (also OHC in line 143) affect TC translation speed? The authors only described the time series of translation speed and position of eddy, but did not clearly demonstrate the related mechanisms. The authors should supply more evidence to demonstrate how the eddy modulates steering flow and then affect TC translation speed.

(2) On the effect of mesoscale eddy on TC intensity change. The authors just described the movement of TC Madi and relative position with respect to the eddies and then concluded the intensity change of Madi was dominated by the eddies. I do know the authors show OHC change during the TC passage, but acctually the key (oceanic) factor controlling TC intensity change is SST. At least, the time series of SST like figures 2-4 should be given to substantiate the eddy effect. Furthermore, the slow TC translation speed may induce large SST cooling and contribute to the weakening of Madi.

(3) On the mechanism of SST and biogeochemical response. The authors concluded that the SST cooling and Cha increase was due to eddy-pumping of subsurface waters. However, there were clear sursurface temperature increase and Cha decrease in the thermocline in Fig. 11, indicating a non-negligible role of diapycnal mixing. This was also consistent with results from many previous studies, i.e., the SST change was mainly due to diapycnal mixing (Price 1981).

Specific comments

(1) line 67: the Unisys Weather does not give TC track information right now. Actually, the TC track information of Unisys Weather is originated from the Joint Typhoon Warning Center.

(2) line 80: The temperature profiles should be indicated to calculate OHC.

(3) line 123-124: Most people may be not familiar with the classification of intensity of IMD. Please give the range of wind speed of different IMD categories or use the more popular Saffir-Simpson scale.

(4) line 149 & 150: Compared with the huge OHC of the ocean, the heat uptake by a TC was very small. The decrease of local OHC may be subject to the advection of TC induced strong currents.

(5) line 189 & 200: To examine the effect of vertical wind shear on TC, people mostly average the vertical wind shear azimuthally around the TC center, not the spatial map as in Fig. 5. Relatively, vertical wind shear of 10-15 m/s is not small and may compromise TC intensification.

Technical corrections

(1) line 25: "occurred" should be "ever reported"

(2) line 153: delete the first "it"

(3) line 339: delete the second "and"

(4) Line 875-883: The line number overlaying the figure legend is confusing.

Reference: Price J F. Upper ocean response to a hurricane[J]. Journal of Physical Oceanography, 1981, 11(2): 153-175.

---

## Referee Comment (RC4) · Anonymous Referee #1 · 28 Mar 2019

I do not think that the authors completely replied on "Role of cold core eddy in controlling / arresting the northward movement of cyclone Madi", particularly "the slow down of the northward movement of cyclone Madi and its final arrest was mediated by the presense of oceanic cyclonic eddy". The parameter "Feddy" could explain only the intensity change of a cyclone such as "positive" or " negative" feedback when a translation speed and oceanic parameters were given.

I would like to argue that the authors need to study the effect of a cold eddy on the movement of a cyclone using another method such as numerical experiments by the

coupled atmosphere-ocean model with/without a cold eddy in order to show evidence. At least, it is unreasonable to conclude the effect of a cold eddy on the cyclone movement only with the data used in this study. Otherwise, the authors could find statistical evidence if they analyze the best track data.

Descriptions of biogeochemical oceanic responses to a cyclone are improved with more quantitative descriptions. However, the authors could not provide evidence for the effects of a cold eddy on the cyclone movement, although the effects of a cold eddy on the cyclone intensity change became clear. Because the limit of the open status is 3rd April, I recommend rejection in the current discussion paper.

---

## Short Comment (SC3) · 30 Mar 2019

Reply to the Comments of Anonymous Referee#2 Posted on 26 March 2019

General Comment

• Referee's General Comment Based a suite of atmospheric and oceanic datasets during the passage of TC Madi, Chowdhury et al. examined the upper ocean physical-biogeochemical response to the TC, mostly emphasized the effect of pre-existing cold core eddies underneath the TC. The topic of TC-ocean interaction in the BoB is interesting and important for TC forecasting. Generally, the effect of mesoscale eddy on TC-ocean interaction is well known at the present stage. Due the lack of in situ observations, studies on the Biogeochemical response to a TC is relatively less and this study may enrich our knowledge on the biogeochemical change induced by TC passage.

Author's Response We thank the Reviewer#2 for reviewing the manuscript and for the comments.

For Authors Response Please Access Supplement pdf

Please also note the supplement to this comment:
https://www.ocean-sci-discuss.net/os-2018-133/os-2018-133-SC3-supplement.pdf

**Supplement:**

Reply to the Comments of Anonymous Referee#2 Posted on 26 March 2019

**General Comment**

**Referee's General Comment**

Based a suite of atmospheric and oceanic datasets during the passage of TC Madi, Chowdhury et al. examined the upper ocean physical-biogeochemical response to the TC, mostly emphasized the effect of pre-existing cold core eddies underneath the TC. The topic of TC-ocean interaction in the BoB is interesting and important for TC forecasting. Generally, the effect of mesoscale eddy on TC-ocean interaction is well known at the present stage. Due the lack of in situ observations, studies on the Biogeochemical response to a TC is relatively less and this study may enrich our knowledge on the biogeochemical change induced by TC passage.

**Author's Response**

We thank the Reviewer#2 for reviewing the manuscript and for the comments.

**• Referee's General Comment**

In the manuscript, I find some conclusions are inaccurate or unclear with not sufficient evidences, especially on the effect of mesoscale eddies. Therefore, I suggest a major revision prior publication. I hope the following comments are useful when the authorsrevise their manuscript.

(1) How does cyclonic eddy (also OHC in line 143) affect TC translation speed? The authors only described the time series of translation speed and position of eddy, but did not clearly demonstrate the related mechanisms. The authors should supply more evidence to demonstrate how the eddy modulates steering flow and then affect TC translation speed.

**Author's Response**

To address the concern of the reviewer about the effect of mesoscale eddies and to demonstrate the role of eddies in modulating the translation speed of cyclone Madi we have used a two-prong approach. First, we calculated the time evolution maps of difference of SST of 5th December (pre-cyclone SST) from each day starting from 6th December to 14th December (See Figure A) to show the large SST cooling in the north (the location of cold core eddy), which led to the weakening of tropical cyclone Madi. Second, to quantify the role of eddy in reducing the speed of northward movement of tropical cyclone Madi in the region

---

## Short Comment (SC4) · 1 Apr 2019

Reply to the Comments of Anonymous Referee#1 Posted on 28 March 2019

Referee's Comment

I do not think that the authors completely replied on "Role of cold core eddy in controlling / arresting the northward movement of cyclone Madi", particularly "the slow down of the northward movement of cyclone Madi and its final arrest was mediated by the presence of oceanic cyclonic eddy". The parameter "Eeddy" could explain only

the intensity change of a cyclone such as "positive" or " negative" feedback when a translation speed and oceanic parameters were given.

Author's Response

We wish to show to the reviewer the 3-dimensional response of the cyclone Madi in terms of SST cooling was a significant factor in Madi's rapid weakening (as also suggested by the eddy feedback factor which showed that the contribution of cyclonic eddy in reducing the storm intensity was 69%) by presenting the time evolution maps of difference of SST of 5th December (pre-cyclone SST) from each day starting from 6th December to 15th December (See Figure A in the previous page). to show the large SST cooling in the north (the location of cold core eddy).

The time evolution of difference in SST from 6th to 10th December showed a distinct cooling of 2 to2.5oC in the region of affected by the cyclone Madi. A comparison of these maps with Fig.3 of the manuscript clearly points that in the northern most region of the cyclone track, where there a cyclonic eddy was pre-existing; the cooling of SST was 2.5oC, which was 0.5oC colder than the rest of the region. The excess cooling of 0.5oC noticed in the eddy region lends support to the notion that the slow translation speed led to the further cooling of SST, which contributed to the weakening of the cyclone from VSCS to SCS, through negative feedback.

Referee's Comment

I would like to argue that the authors need to study the effect of a cold eddy on the movement of a cyclone using another method such as numerical experiments by the coupled atmosphere-ocean model with/without a cold eddy in order to show evidence. At least, it is unreasonable to conclude the effect of a cold eddy on the cyclone movement only with the data used in this study. Otherwise, the authors could find statistical evidence if they analyze the best track data.

Author's Response

The Reviewer's suggestion of numerical experiment to study the effect of cold eddy on the movement of cyclone is welcome, but it is beyond the scope of our present paper. As indicated in our manuscript at line 355, we recognise the lack of modelling studies as one of our limitation, which we intend to carryout in near future.

We beg to disagree with the reviewer that "it is unreasonable to conclude the effect of a cold eddy on cyclone movement only with the data used in this study". We have used all possible data, both in situ as well as remote sensing, and argued our case at a reasonable level.

Referee's Comment

Descriptions of biogeochemical oceanic responses to a cyclone are improved with more quantitative descriptions. However, the authors could not provide evidence for the effects of a cold eddy on the cyclone movement, although the effects of a cold eddy on the cyclone intensity change became clear. Because the limit of the open status is 3rd April, I recommend rejection in the current discussion paper.

Author's Response

We have used all possible data, both in situ as well as remote sensing, and argued our case at a reasonable level. In spite of this, if the Reviewer wants to turn down our study just because it is only based on data analysis, is unfortunate.

Complete Response to Reviewer 1 is in Supplement pdf

Please also note the supplement to this comment:
https://www.ocean-sci-discuss.net/os-2018-133/os-2018-133-SC4-supplement.pdf

———————————————————

Reply to the Comments of Anonymous Referee#1 Posted on 28 March 2019

[Figure]

**Figure A**. Time evolution maps of difference of SST of 5th December (pre-cyclone SST) from each day starting from 6th December to 15th December.

**Fig. 1.** Figure A

---

## Author Comment (AC1) · 16 Apr 2019

**Reply to the Comments of Anonymous Referee #1** 1 2 [Received and Published: 31 January 2019] 3 4 1. **General comments:** 5 6 • **Referee's Comment** This paper studies the upper ocean physical and biogeochemical response to Cyclone Madi 7 (2013) in the Bay of Bengal from the analysis of multiple atmospheric and oceanic data. The 8 topic is interesting. My understanding is that there are many studies regarding the topic: the 9 upper ocean physical and biogeochemical response to cyclones. Some of the studies have 10 been referenced in this manuscript. Therefore, my concern is the novelty of this paper. 11 12 **Author's Response** 13 • 14 In this paper we not only bring out the cyclone-induced physical and biogeochemical response of the surface ocean using a suite of remote sensing data, we document the time-15 evolving depth-dependent temperature and chlorophyll response of the upper ocean to 16 tropical cyclone, which is the novelty of this paper. 17 18 We have used two Argo float data (WMO ID 2901288, 2901629) for temperature profiles 19 20 that were in the vicinity of Track 2 and one Bio-Argo float (WMO ID 2902086) for 21 chlorophyll profiles that was to the right of Track 1. 22 23 Another novelty is that the enhancement of Chl-a and net primary productivity by cyclone Madi is much greater compared with previous cyclones that occurred in the BoB. We show 24 that this is the combined impact of cyclone and oceanic cyclonic eddy. This is significant 25 because BoB is known for its low upper ocean Chl-a and primary productivity. 26 27 28 Authors' Changes in Manuscript • 29 Following text will be added to the original ms at line 357: 30 31 32 The time-evolving depth-dependent temperature and chlorophyll response of the upper ocean to tropical cyclone and greater enhancement of Chl-a and net primary productivity by cyclone 33 34 Madi compared with previous cyclones that occurred in the BoB are the novelty of this paper.

35 36

**2. Referee's Comment**

37

There are a few discussion points on this study.

1) I don't agree with the conclusion that "oceanic eddies affect the translation speed of
Madi". This study lacks discussions on what factors do affect the movement of cyclones. In
addition, there is no information regarding the atmospheric steering flow and wave-1
asymmetry of the inner core structure that is very important in understanding the movement

of storms, although the authors showed the horizontal distributions of 500-hPa winds in Fig.
6. There is a cause-result confusion on the authors' conclusion that the motion of the cyclone
is mainly controlled by cold eddy when the cyclone stagnated over the cold eddy without any
considerations of the variation of environmental steering flow.

48 49

**Author's Response**

(1)

To address the concern of the reviewer, especially the role of atmospheric steering, we wishto clarify this point with the help of additional computations and figures.

52

Environmental conditions that are necessary for the formation of tropical cyclones are (1)
SST in excess of 26.5°C, (2) low level wind convergence, (3) high relative vorticity, (4) weak
vertical wind shear, and (5) high relative humidity.

56

57 Once formed cyclones generally move in a westerly/north-westerly direction due to the 58 advection of potential vorticity field by storm circulation. In simple case, the background 59 potential vorticity gradient is simply the meridional gradient of the Coriolis parameter (f) 60 where  $f = 2\Omega \sin \Phi$  and  $\Phi$  varies with the latitude, beta ( $\beta = \frac{\partial f}{\partial y}$ ). The Beta drift generally 61 causes the tropical cyclone to move poleward and westward relative to the motion they would 62 have if the background potential vorticity field were unperturbed by the storms.

Based on the comments by the reviewer, in order to examine the role of environmental steering flow, we show the analysis of the vorticity dynamics, which is the tendency of the absolute vorticity term  $(\frac{\partial \eta}{\partial t})$ , as per the following equation:

67

$$\qquad \frac{\partial \eta}{\partial t} = -\vec{v} \cdot \nabla \left(\zeta + f\right) - \left(\zeta + f\right) \left(\nabla \cdot \vec{v}\right) - w \frac{\partial \varepsilon}{\partial p} + \left(\frac{\partial w}{\partial x} \frac{\partial v}{\partial z} - \frac{\partial w}{\partial y} \frac{\partial u}{\partial z}\right) + \vec{F}$$

(2)

69

70 Where, u,v,w are the velocity component. The vertical component  $\eta$  of the absolute vorticity 71 vector (as defined above) given by the sum of the vertical component of vorticity with respect 72 to the earth (relative vorticity ( $\zeta$ )) and the vorticity of the earth (equal to the Coriolis 73 parameter *f*).

(4)

(5)

(3)

74

75  $\zeta = \frac{\partial v}{\partial x} - \frac{\partial u}{\partial y}$  where *u* and *v* are zonal and meridional wind.

76  $f = 2\Omega \sin \Phi$  where  $\Phi$  varies with the latitude, and  $\eta = \zeta + f$ 77

The local change of absolute vorticity is related to horizontal advection (term 1), stretching term (term 2; divergence multiplied by absolute vorticity), vertical advection (term 3), tilting term (term 4)and friction (term 5). The contribution from the each term in the absolute vorticity tendency equation was examined. The analysis has shown that at lower level (850hpa) stretching and horizontal advection in the vorticity equation are the major contributors of the absolute vorticity tendency term. The contribution from other terms being very small could be neglected.

Low level vorticity being one of the environmental factor which plays a key role in the formation and development of the tropical cyclone, the time evolution of vorticity tendency (Fig.A), horizontal advection (Fig.B) and stretching term (Fig.C), all at 850 hpa, were analyzed.

---

## Author Comment (AC2) · 16 Apr 2019

| 1      | Reply to the Comments of Anonymous Referee #1                                                                                                                                         |
|--------|----------------------------------------------------------------------------------------------------------------------------------------------------------------------------------------------|
| 2      | [Received and Published: 11 March 2019]                                                                                                                                                      |
| 3      |                                                                                                                                                                                              |
| 4
| 1. General comments:                                                                                                                                                                         |
| 6
| Referee's Comment                                                                                                                                                                            |
| 8
| I am grateful for the authors' replies. Although the presentation of the paper might have been improved, I can not confirm it at the present moment, because I cannot find the revised paper |

in the web now. Therefore, my comments are only for the replies.

11

**Author's Response**

We have gone through the comments of the Referee#1 very carefully. In fact, there are 4 14 comments this time and we addressed them point-by-point in the following section. However, 15 all the 4 points broadly converge to Referee's reviews about (1) role of cold core eddy in 16 controlling/arresting the northward movement of cyclone Madi and (2) how the present study 17 on the out-gassing of CO2 is quantitatively different from that of earlier studies.

**18 (1) Role of cold core eddy in controlling/arresting the northward movement of cyclone Madi**

It is unfortunate that Referee#1 still remains unconvinced about the role of cold core eddy in 19 controlling the northward movement of the cyclone. 20

To quantify the eddy's contribution to the intensity of the cyclone Madi, we have calculated the eddy feedback factor following Wu et al. (2007). The analysis showed (for details see 22 following section on Reply to Reviewer's specific comments) that from 7th to 8th December 23 when the system intensified from CS to VSCS and was passing through the warm patch 24 associated with warm core eddy (see spatial maps of OHC at Fig.2 & positive SLA at Fig.3 25 of original manuscript) the eddy feedback factor was positive and amounted to 59%. 26 Thereafter, when the cyclone passed over the cold patch associated with cold core eddy 27 during 9th and 10th December, the eddy feedback factor was negative and 69%. Thus, this 28 analysis quantifies the contribution of both warm and cold core eddies; when cyclone passed 29 through the warm patch the system intensified from CS to VSCS and its translation speed 30 increased (see Table 1), while when it passed over the cold patch from 9thto 10th the system 31 slowed down and its northward movement was arrested as noted under the section 2.3. We 32 have elaborated the methodology of computation of eddy feedback factor under the 33 "Response to specific comments" with a new diagram. The modification to the manuscript is 34 also indicated there. We hope this will convince the Reviewer#1 35

**37 2) How the present study on the out-gassing of $CO_2$ is quantitatively different from that of 38 earlier studies**

Based on the Reviewer's comment, in order to compare our CO2 flux with that of previous studies, we have recomputed the CO2 flux along Track 1, Track 2 and Boxes A and B in 40 mmol per meter square per day. While re-computing CO2 flux we noticed a bug in our 41 previous calculation, which we rectified. The newly calculated values showed a cyclone-42 induced CO2 out-gassing which was about 4-times greater than the pre-cyclone values along 43 44 Track 1 and in Box B. The impact of CO2 out-gassing in Box A and along Track 2 were much smaller as when the cyclone was in this box and was passing through this track it was 45 46 in a formative and dissipative stage respectively.

Several studies have demonstrated that the passage of a tropical cyclone can lead to enormous 47 48 amount of CO2 flux from the ocean surface to the atmosphere. For example, based on observation from Sargasso Sea during summer 1995 Bates et al. (1998) showed that 49 hurricanes accounted for nearly 55% of the CO2 flux into the atmosphere, while based on 50 moored buoy data from the East China Sea Nemoto et al. (2009) reported a 60% contribution 51 from typhoon in summer. In the eastern Arabian Sea Byju and Prasanna Kumar (2011) noted 52 that cyclone Phyan emitted ~8 mmol  $m^2$  day1 of CO2 from ocean to atmosphere accounting 53 54 for  $\sim 85\%$  of the total out-gassing for the month of November (climatology) calculated by Takahashi et al. (2009). Our study show that during cyclone Madi (6-12 Dec) Maximum CO2 55 flux observed was 13 mmol m-2 day-1. Tropical cyclones have significant impact on the 56 carbon cycle in the Bay of Bengal (Ye et al., 2019). Based on their study cyclone Hudhud and 57 cyclone Roanu formed over the Bay of Bengal enhanced CO2 efflux (18.49  $\pm$  3.70 mmol 58  $CO_2 \text{ m}^{-2} \text{ day}^{-1}$ ) and  $(19.08 \pm 3.82 \text{ mmol } CO_2 \text{ m}^{-2} \text{ day}^{-1})$  due to wind effect during the storm. 59

We have elaborated this under Response to specific comments with a new diagram. The 60 modification to the manuscript is also indicated there. 61

63

- Authors' Changes in Manuscript •
- 65 No change in the manuscript in response to this query.
- 66
- 67 68

2. **Referee's Specific Comment**

For the biogeochemical oceanic response to a storm, I understand that the observation itself is 70 new. However, the authors did not reply (and revise) the following scientific themes: What 71 72 kind of processes did the oceanic response occur by? What effects did the response have on?. 73 74 75

**Author's Response**

Sorry to say that we did not quite understand the question of the Reviewer. Assuming that the Reviewer is enquiring about the oceanic processes that are responsible for the response in terms of enhanced chlorophyll concentration and enhanced  $CO_2$  out-gassing, following is our reply.

As the tropical cyclone Madi passes over the BoB, the upper ocean experiences strong vertical mixing associated with strong winds, which is essentially a one-dimensional 83 84 response. In addition to this, the cyclonic winds lead to strong Ekman divergence, which is a 85 three-dimensional response. This, in turn, forces the subsurface cold and nutrient rich waters to come to the surface under the upward Ekman pumping. Increased availability of nutrients 86 87 to the upper ocean will initiate the carbon fixation by phytoplankton in the euphotic zone and 88 results in the enhancement in chlorophyll biomass. In addition to the enhancement of 89 chlorophyll biomass with time, the upward Ekman pumping also would result in an increase 90 in the  $CO_2$  out-gassing in the following manner. As the cold subsurface waters comes to the 91 surface it also brings with it higher concentration of dissolved  $CO_2$ . Once at the surface, the 92 warmer temperature and strong winds will initiate a strong out-gassing of  $CO_2$  from ocean 93 surface to the atmosphere. This happens under the action of all tropical cyclone. What is 94 distinct in our case is that the cyclone Madi is passing over cold core eddy. Under this condition the upward transport of CO2 rich subsurface water occurs due to both Ekman 95 96 pumping driven by cyclonic winds associated with the cyclone Madi as well as eddy-97 pumping driven by cyclonic circulation of water in a cold core eddy. Accordingly, in our 98 study we see a nearly 4-fold increase in the  $CO_2$  out-gassing compare to its pre-cyclone 99 values, when the cyclone passes over track 1 and Box B which has cyclonic eddy. See more 100 details under the next reply.

102

**• Authors' Changes in Manuscript**

- 104 No change in the manuscript in response to this query.
- 105

107

**3. Referee's Specific Comment**

I also understand that there was a sudden change in the biogeochemical components such as Chl-a and outgassing of CO2 from the background before the passage. However, the authors have not shown any evidence of the difference quantitatively from the present study to Bate et al. (1998) and Nemoto et al. (2009). According to Wada et al. (2011), the amount of outgassing of CO2 is greatly affected by the error of surface wind speed analysis data. Therefore, I think that the difference the authors found is not a new finding but the result including the observational error.

- 116
- 117 Author's Response

The air-sea CO2 flux at the sea surface depend on the difference between the partial pressure 118 of  $CO_2$  at the sea surface (p $CO_2^{sea}$ ) and in the overlying atmosphere (p $CO_2^{air}$ ), the wind 119 speed, sea surface temperature and sea surface salinity as per the equations 4, 5 and 6 in the 120 manuscript. Among these factors, the wind speed plays an important role in determining the 121 value of air-sea  $CO_2$  flux due to the quadratic functional dependence of the gas transfer 122 velocity with the wind speed. Several studies have demonstrated that the passage of a tropical 123 124 cyclone can lead to enormous amount of  $CO_2$  flux from the ocean surface to the atmosphere. For example, based on observation from Sargasso Sea during summer 1995 Bates et al. 125 (1998) showed that hurricanes accounted for nearly 55% of the CO2 flux into the atmosphere, 126 while based on moored buoy data from the East China Sea Nemoto et al. (2009) reported a 127 60% contribution from typhoon in summer. In the eastern Arabian Sea Byju and Prasanna 128 Kumar (2011) noted that cyclone Phyan emitted ~8 mmol m2 day1 of CO2 from ocean to 129 atmosphere accounting for ~85% of the total out-gassing for the month of November 130 (climatology) calculated by Takahashi et al. (2009). Our study show that during cyclone 131 Madi (6-12 Dec) maximum CO2 flux observed was 13.73 mmol m-2 day-1. Tropical cyclones 132 have significant impact on the carbon cycle in the Bay of Bengal (Ye et al., 2019). Based on 133 134 their study cyclone Hudhud and cyclone Roanu formed over the Bay of Bengal enhanced CO2 efflux  $(18.49 \pm 3.70 \text{ mmol CO}_2 \text{ m}^{-2} \text{ day}^{-1})$  and  $(19.08 \pm 3.82 \text{ mmol CO}_2 \text{ m}^{-2} \text{ day}^{-1})$  due 135 to wind effect during the storm. 136

Based on the Reviewer's comment in order to compare our  $CO_2$  flux with that of previous studies we have recomputed the  $CO_2$  flux along Track 1, Track 2 and Boxes A and B in mmol per meter square per day. While re-computing  $CO_2$  flux we noticed a bug in our previous calculation, which we rectified and a new figure is generated as given below.

**152** Fig A. Daily variation total  $CO_2$  flux (mmol/m2/day) in the Box A (red) and B (blue) and along Track 1 (green)

and 2 (black) from 2 to 15 December 2013. The vertical lines are the standard deviations.

The newly calculated values showed a cyclone-induced  $CO_2$  out-gassing which was about 4 and 4-times greater than the pre-cyclone values along Track 1 and Box B respectively. The impact of  $CO_2$  out-gassing in Box A and along Track 2 were much smaller because when the cyclone was in Box A and passing through Track 2 it was in a formative and dissipative stages respectively.

Regarding the observational error associated with wind data used in our present study, the bias and root-mean-square differences of the wind speed between ASCAT and dropwindsonde data are -1.7 and 5.3 ms-1 (Chou et al., 2013). ASCAT winds are most reliable when the wind speeds are in the range of 12 and 18 m s-1 and can be applied to determine the radius of 34 knot winds, a critical parameter in operational tropical cyclone analysis (Chou et al., 2013).

In our case the wind speed used for the computation of  $CO_2$  flux ranged from 2.72 to 10.38 166 ms-1. We have calculated the correlation between Rama buoy and ASCAT wind data in BoB 167 region. We have chosen wind speed data from the RAMA buoy located at 12 N 90 E and 15 168 169 N 90 E for the comparison with ASCAT wind product which is used for the CO2 flux calculation in our study. The location of the Rama buoy was nearby the track of the cyclone 170 171 taken from Global Madi. Rama data was tropical moored buov arrav 172 (https://www.pmel.noaa.gov/tao/drupal/disdel/). The correlation Coefficient values are 0.89 and 0.83 respectively indicating the quality of the ASCAT data 173

175

Fig B. Comparison between wind data from RAMA buoy and ASCAT wind data for December 2013

**• Authors' Changes in Manuscript**

Following text will be added to the original ms at line no. 327:

Consistent with Chl-*a* and NPP, the maximum  $CO_2$  out-gassing to the atmosphere was seen in 183 Box B and along Track 1. In Box B, the maximum  $CO_2$  out-gassing was  $13.73 \pm 2.47$  mmol 184  $m^{-2} day^{-1}$  to the atmosphere which was nearly 4-fold higher its average pre-cyclone value 185  $(3.50 \pm 0.07 \text{ mmol } m^{-2} day^{-1})$ . The maximum CO2 out-gassing along Track 1 was 13.22  $\pm$ 186 2.50 mmol  $m^{-2} day^{-1}$  to the atmosphere which was nearly 3-fold higher its average pre-187 cyclone  $(4.59 \pm 0.20 \text{ mmol } m^{-2} day^{-1})$ . The impact of cyclone Madi on CO2 out-gassing in 188 Box A and along Track 2 were much smaller due to the less wind effect as the cyclone was in 189 a formative and dissipative stage respectively.

We will also replace the Fig.14 with the following new Figure (Fig.16) at line no. 968 of the original manuscript.

---

## Author Comment (AC3) · 16 Apr 2019

**[Received and Published: 28 March 2019]**

**1.        Referee's Comment**

I do not think that the authors completely replied on "Role of cold core eddy in controlling / arresting the northward movement of cyclone Madi", particularly "the slow down of the northward movement of cyclone Madi and its final arrest was mediated by the presense of oceanic cyclonic eddy". The parameter "Feddy" could explain only the intensity change of a cyclone such as "positive" or " negative" feedback when a translation speed and oceanic parameters were given.

- **Author's Response**

We wish to show to the reviewer the 3-dimensional response of the cyclone Madi in terms of SST cooling was a significant factor in Madi's rapid weakening (as also suggested by the eddy feedback factor which showed that the contribution of cyclonic eddy in reducing the storm intensity was 69%) by presenting the time evolution maps of difference of SST of 5th December (pre-cyclone SST) from each day starting from 6th December to 15th December (See Figure A in the previous page). to show the large SST cooling in the north (the location of cold core eddy).

[Figure]

**Figure A**. Time evolution maps of difference of SST of 5th December (pre-cyclone SST) from each day starting from 6th December to 15th December.

The time evolution of difference in SST from 6$^{th}$ to 10$^{th}$ December showed a distinct cooling of 2 to2.5$^{o}$C in the region of  affected by the cyclone Madi. A comparison of these maps with Fig.3 of the manuscript clearly points that in the northern most region of the cyclone track, where there a cyclonic eddy was pre-existing; the cooling of SST was 2.5$^{o}$C, which was 0.5$^{o}$C colder than the rest of the region. The excess cooling of 0.5$^{o}$C noticed in the eddy region lends support to the notion that the slow translation speed led to the further cooling of SST, which contributed to the weakening of the cyclone from VSCS to SCS, through negative feedback.

- **Authors' Changes in Manuscript**

No change in the manuscript in response to this query.

**2.     Referee's Comment**

I would like to argue that the authors need to study the effect of a cold eddy on the movement of a cyclone using another method such as numerical experiments by the coupled atmosphere-ocean model with/without a cold eddy in order to show evidence.
At least, it is unreasonable to conclude the effect of a cold eddy on the cyclone movement only with the data used in this study. Otherwise, the authors could find statistical evidence if they analyze the best track data.

- **Author's Response**

The Reviewer's suggestion of numerical experiment to study the effect of cold eddy on the movement of cyclone is welcome, but it is beyond the scope of our present paper. As indicated in our manuscript at line 355, we recognise the lack of modelling studies as one of our limitation, which we intend to carryout in near future.

We beg to disagree with the reviewer that "it is unreasonable to conclude the effect of a cold eddy on cyclone movement only with the data used in this study". We have used all possible data, both in situ as well as remote sensing, and argued our case at a reasonable level.

- **Authors' Changes in Manuscript**

No change in the manuscript in response to this query.

**3.     Referee's Comment**

Descriptions of biogeochemical oceanic responses to a cyclone are improved with
more quantitative descriptions. However, the authors could not provide evidence for
the effects of a cold eddy on the cyclone movement, although the effects of a cold
eddy on the cyclone intensity change became clear. Because the limit of the open
status is 3rd April, I recommend rejection in the current discussion paper.

**Author's Response**

We have used all possible data, both in situ as well as remote sensing, and argued our case at
a reasonable level.

In spite of this, if the Reviewer wants to turn down our study just because it is only based on
data analysis, is unfortunate.

- **Authors' Changes in Manuscript**

No change in the manuscript in response to this query.

---

## Author Comment (AC4) · 16 Apr 2019

| 1                                                              | Reply to the Comments of Anonymous Referee #2                                                                                                                                                                                                                                                                                                                                                                                                                                                                                                                                                                                                                               |
|----------------------------------------------------------------|------------------------------------------------------------------------------------------------------------------------------------------------------------------------------------------------------------------------------------------------------------------------------------------------------------------------------------------------------------------------------------------------------------------------------------------------------------------------------------------------------------------------------------------------------------------------------------------------------------------------------------------------------------------------------------|
| 2                                                              | [Received and Published: 26 March 2019]                                                                                                                                                                                                                                                                                                                                                                                                                                                                                                                                                                                                                                            |
| 3                                                              |                                                                                                                                                                                                                                                                                                                                                                                                                                                                                                                                                                                                                                                                                    |
| 4
                                     | <ol> <li>General comments:</li> <li>Referee's Comment</li> </ol>                                                                                                                                                                                                                                                                                                                                                                                                                                                                                                                                                                                                                   |
| 8
| Based a suite of atmospheric and oceanic datasets during the passage of TC Madi,
Chowdhury et al. examined the upper ocean physical-biogeochemical response to the TC,
mostly emphasized the effect of pre-existing cold core eddies underneath the TC. The topic
of TC-ocean interaction in the BoB is interesting and important for TC forecasting.
Generally, the effect of mesoscale eddy on TC-ocean interaction is well known at the present
stage. Due the lack of in situ observations, studies on the Biogeochemical response to a TC is
relatively less and this study may enrich our knowledge on the biogeochemical change
induced by TC passage. |
| 18                                                             | Author's Response                                                                                                                                                                                                                                                                                                                                                                                                                                                                                                                                                                                                                                                                  |
| 19                                                             | We thank the Reviewer#2 for reviewing the manuscript and for the comments.                                                                                                                                                                                                                                                                                                                                                                                                                                                                                                                                                                                                         |
| 20                                                             |                                                                                                                                                                                                                                                                                                                                                                                                                                                                                                                                                                                                                                                                                    |
| 21
                                       | • Authors' Changes in Manuscript                                                                                                                                                                                                                                                                                                                                                                                                                                                                                                                                                                                                                                                   |
| 24
                                 | No change in the manuscript in response to this query.                                                                                                                                                                                                                                                                                                                                                                                                                                                                                                                                                                                                                             |
| 28                                                             | 2. Referee's General Comment                                                                                                                                                                                                                                                                                                                                                                                                                                                                                                                                                                                                                                                       |
| 29
| In the manuscript, I find some conclusions are inaccurate or unclear with not sufficient evidences, especially on the effect of mesoscale eddies. Therefore, I suggest a major revision prior publication. I hope the following comments are useful when the authors revise their manuscript.
(1) How does cyclonic eddy (also OHC in line 143) affect TC translation speed? The authors only described the time series of translation speed and position of eddy, but did not clearly demonstrate the related mechanisms. The authors should supply more evidence to demonstrate how the eddy modulates steering flow and then affect TC translation speed.                    |

• Author's Response

To address the concern of the reviewer about the effect of mesoscale eddies and to 41 42 demonstrate the role of eddies in modulating the translation speed of cyclone Madi we have used a two-prong approach. First, we calculated the time evolution maps of difference of SST 43 of 5th December (pre-cyclone SST) from each day starting from 6th December to 14th 44 December (See Figure A) to show the large SST cooling in the north (the location of cold 45 core eddy), which led to the weakening of tropical cyclone Madi. Second, to quantify the role 46 of eddy in reducing the speed of northward movement of tropical cyclone Madi in the region 47 of eddy we have calculated the eddy feedback factor following Wu et al. (2007) which is 48 49 presented in Figure B.